# DECO: Decoupled Multimodal Diffusion Transformer for Bimanual Dexterous Manipulation with a Plugin Tactile Adapter

Xukun Li [* 1 2 3]  Yu Sun [* ‡ 2]  Lei Zhang [2 4]  Bosheng Huang [2 5]  Yibo Peng [2]  Yuan Meng [3]  Haojun Jiang [5]
Shaoxuan Xie [2]  Guocai Yao [2]  Alois Knoll [3]  Zhenshan Bing [† 6]  Xinlong Wang [† 2]  Zhenguo Sun [† 1 2 3]

Code: https://github.com/BAAI-Humanoid/DECO
Project Page: https://baai-humanoid.github.io/DECO-webpage/
Dataset: https://huggingface.co/datasets/BAAI-Humanoid/DECO-50

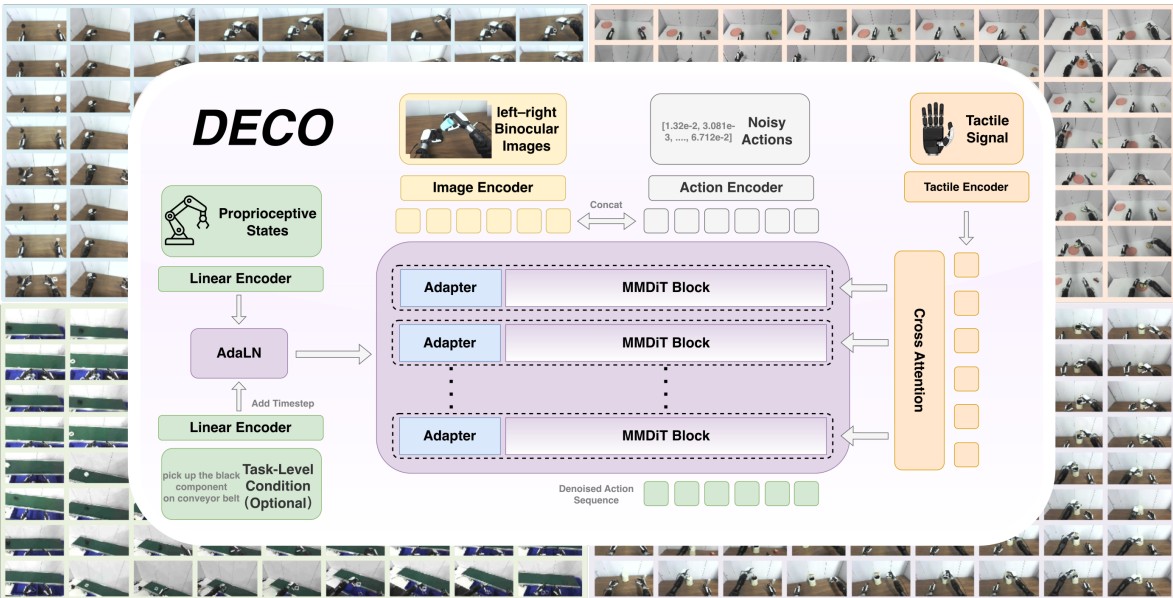

*Figure 1.* **Overview of the Proposed DECO Framework.** DECO is a DiT-based policy that decouples multimodal conditioning. Image and action tokens interact via joint self attention, while proprioceptive states and optional conditions are injected through adaptive layer normalization. Tactile signals are injected via cross attention, while a lightweight LoRA-based adapter is used to efficiently fine-tune the pretrained policy. DECO is also accompanied by DECO-50, a bimanual dexterous manipulation dataset with tactile sensing, consisting of 4 scenarios and 28 sub-tasks, covering more than 50 hours of data, approximately 5 million frames, and 8,000 successful trajectories.

---

[*]Equal contribution [‡]Project lead [†]Corresponding author [1]XYZ Embodied AI, Beijing, China [2]Beijing Academy of Artificial Intelligence, Beijing, China [3]School of Computation, Information and Technology, Technical University of Munich, Garching, Germany. [4]Department of Shenyang Institute of Computing Technology, University of Chinese Academy of Sciences, Beijing, China [5]Department of Computer Science and Technology, Tsinghua University, Beijing, China. [6]State Key Laboratory for Novel Software Technology, Nanjing University, Nanjing, China. Correspondence to: Zhenshan Bing, Xinlong Wang, Zhenguo Sun <hitsunzhenguo@gmail.com>.

*Proceedings of the $43^{rd}$ International Conference on Machine Learning*, Seoul, South Korea. PMLR 306, 2026. Copyright 2026 by the author(s).

## Abstract

Bimanual dexterous manipulation relies on integrating multimodal inputs to perform complex real-world tasks. To address the challenges of effectively combining these modalities, we propose DECO, a decoupled multimodal diffusion transformer that disentangles vision, proprioception, and tactile signals through specialized conditioning pathways, enabling structured and controllable integration of multimodal inputs, with a lightweight adapter for parameter-efficient injection of additional signals. Alongside DECO, we release DECO-50 dataset for bimanual dex-

terous manipulation with tactile sensing, consisting of 50 hours of data and over 5M frames, collected via teleoperation on real dual-arm robots. We train DECO on DECO-50 and conduct extensive real-world evaluation with over 2,000 robot rollouts. Experimental results show that DECO achieves the best performance across all tasks, with a 72.25% average success rate and a 21% improvement over the baseline. Moreover, the tactile adapter brings an additional 10.25% average success rate across all tasks and a 20% gain on complex contact-rich tasks while tuning less than 10% of the model parameters.

## 1. Introduction

Bimanual manipulation is fundamental to human daily life, enabling complex tasks that require coordination between both hands. Extensive research has explored it in daily scenarios (Zheng et al., 2025) and industrial applications (Hou et al., 2025). Recent advances in manipulation policies—from large-scale VLAs to diffusion-based models (Brohan et al., 2023; Liu et al., 2025; Kim et al., 2024; Chi et al., 2025)—have shown rapid progress. Yet hardware limits, particularly rigid grippers that physically interact with objects, constrain dexterity (An et al., 2026; Li et al., 2025). In contrast, humans rely on dexterous hands with rich tactile feedback and fine-grained motor control. To bridge this gap, the community has increasingly focused on dexterous robotic hands that better mimic these capabilities.

Dexterous manipulation policies that rely on vision and proprioception have made considerable progress (Luo et al., 2025; 2026), achieving strong performance on many tasks that do not heavily depend on tactile signal. While some studies have begun integrating tactile signal into dexterous manipulation (Heng et al., 2025), how tactile information is integrated often remains straightforward or underexplored. Moreover, in such visuo-tactile policies, modalities are typically fused in a coupled manner with equal importance (Lin et al., 2025; Cheng et al., 2025), which may fail to fully exploit the distinct roles of vision, proprioception, and touch for action generation—especially when vision changes rapidly with an active camera and tactile signals remain sparse during manipulation. Taken together, these considerations motivate a decoupled paradigm with modality-specific injection.

To address these challenges, we propose **DECO**, a **DEC**oupled multim**O**dal Diffusion Transformer (DiT) paradigm that decouples the conditional injection of visual, proprioceptive states, and tactile modalities. Built on this paradigm, we introduce a **plugin tactile adapter** to inject tactile information into pretrained DECO and improve performance on contact-rich tasks while tuning fewer than 10% of the parameters. We also release **DECO-50**, a bimanual dexterous manipulation dataset with tactile and active vision, and train DECO on it.

In summary, our contributions are threefold:

- **DECO:** We present a decoupled multimodal DiT that conditions on visual, proprioceptive states, and tactile modalities via separate, decoupled injections. Under the assumption that different modalities contribute differently to action generation, this design improves policy performance over coupled fusion.

- **Tactile Adapter:** We propose a plugin tactile adapter that significantly improves vision-based DECO on contact-rich tasks by training only a small fraction of the parameters, demonstrating that tactile can be effectively added to pretrained visuomotor policies.

- **DECO-50:** We release a bimanual dexterous manipulation dataset with tactile and active vision, comprising 4 scenarios, 28 sub-tasks, over 8K successful trajectories and 5M frames. It includes both contact-rich tasks that benefit from tactile and tasks where visual and proprioceptive information suffice.

## 2. Related Work

**Vision-Language-Action Models** Vision-Language-Action (VLA) models have recently demonstrated strong capability and generalization in robot manipulation. Representative works such as RT-1 (Brohan et al., 2023), RDT-1B (Liu et al., 2025), OpenVLA (Kim et al., 2024), and $\pi_0$ (Black et al., 2026) achieve impressive generalization by conditioning policies on language instructions and visual observations. However, in contact-rich settings or under severe visual occlusions, vision alone often fails to reliably capture fine-grained interaction states, where force-related signals are critical for stable and precise manipulation. To enhance vision-based VLA systems, several studies incorporate joint torque (Zhang et al., 2025b), end-effector force/torque (Yu et al., 2025), and tactile information (Huang et al., 2025b; Zhang et al., 2025a) into the VLA framework, improving robustness in tasks such as insertion and assembly. Nevertheless, how to inject multimodal conditions (especially tactile cues) into pretrained policies in a controllable and parameter-efficient manner, while avoiding interference among modalities, remains an open challenge.

**Tactile-Based Policies** With rapid progress in tactile sensor technologies (Huang et al., 2025a; Shan et al., 2025; Lambeta et al., 2024) and multimodal learning, an increasing number of works have explored incorporating tactile information into robotic systems. Most existing works focus on learning unified tactile representations and validate

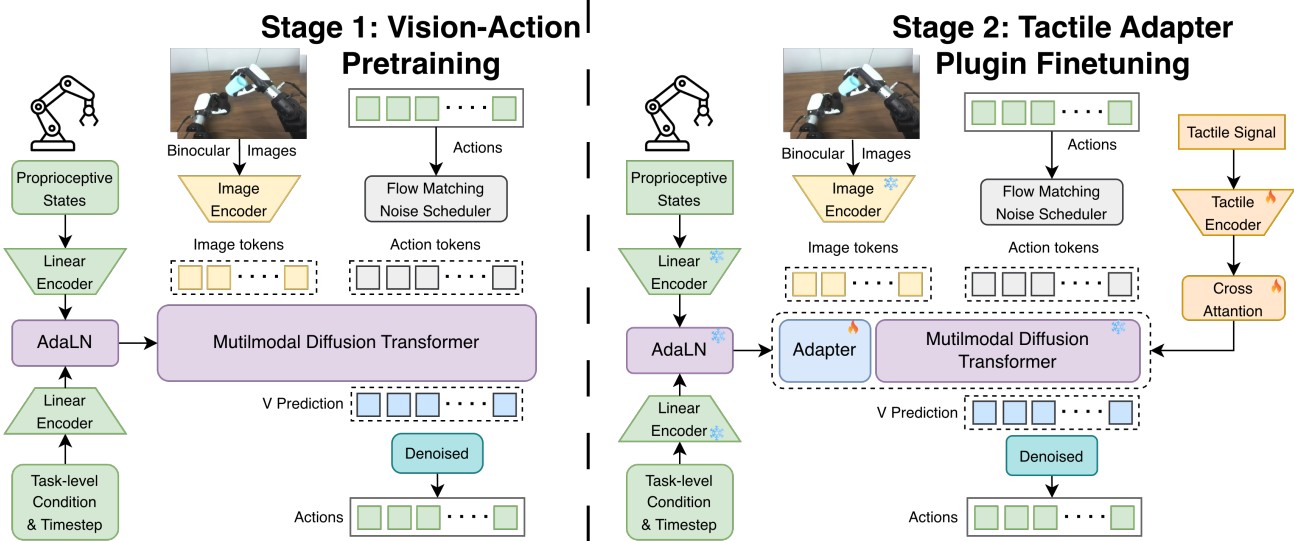

*Figure 2.* **Two-Stage Training Paradigm for DECO.** In the first stage, a vision–action policy is trained with images, proprioceptive states and task-level conditions. In the second stage, the pretrained policy is frozen, and tactile signals are incorporated via a lightweight adapter and cross attention, enabling parameter-efficient adaptation to tactile-aware manipulation without retraining the entire model.

them on downstream tasks such as material classification, cross-modal retrieval, and object reconstruction (Wu et al., 2025d; Dave et al., 2024; Yang et al., 2024; Kerr et al., 2023; Fu et al., 2024; Gao et al., 2022a;b). Some extend to manipulation-related downstream tasks (Yu et al., 2024; Higuera et al., 2025); notably, (Higuera et al., 2025) learns general-purpose touch representations via large-scale self-supervised pretraining and introduces an adapter to integrate tactile cues into downstream policies, yet requires a large ControlNet-based adapter and extensive pretraining data. Other works directly tackle manipulation by jointly predicting actions and tactile signals (Heng et al., 2025), or fusing visual and tactile streams with slow–fast temporal mechanisms (Xue et al., 2025). However, how to effectively and efficiently fuse tactile with other modalities for robot action generation remains insufficiently explored, and there is still a need for fusion schemes that are both effective and parameter-efficient. This motivates more structured, decoupled mechanisms to integrate tactile cues into pretrained policies with low adaptation cost.

**Bimanual Manipulation Datasets** Most existing dexterous manipulation datasets are dominated by single-arm manipulation or static grasp generation, which limits their applicability to complex bimanual tasks. Large-scale real-robot datasets such as DROID (Khazatsky et al., 2024) and RH20T (Fang et al., 2024) offer broad coverage, but they mainly focus on single-arm or mobile manipulation rather than structured bimanual coordination.

Recent benchmarks start to target bimanual manipulation

and cross-embodiment generalization explicitly. Several large-scale datasets focus on bimanual manipulation with grippers, including RoboMIND (Wu et al., 2025a), Robo-MIND 2.0 (Hou et al., 2025), RoboCOIN (Wu et al., 2025c). Moving beyond grippers, datasets for bimanual dexterous manipulation have emerged, such as ActionNet (Fourier ActionNet Team, 2025) and UniHand-2.0 (Luo et al., 2026), which demonstrate the feasibility of learning dexterous bimanual skills from vision and proprioception alone. However, datasets that simultaneously provide bimanual dexterous manipulation trajectories with tactile sensing remain scarce. Addressing these limitations, our work considers bimanual dexterous manipulation tasks with tactile sensing for complex contact-rich tasks.

## 3. Method

DECO follows a two-stage training paradigm: first learning a vision–based policy, and then extending it with tactile sensing via a lightweight adapter while keeping the pretrained policy frozen, as shown in Figure 2. Building upon this training paradigm, we next focus on the model design that enables effective multimodal integration. We first introduce the core block of the DECO framework: Multimodal Diffusion Transformer (MMDiT), which fuses visual, proprioceptive, tactile, and action-related information through modality-specific conditioning mechanisms. This design enables stable and scalable action modeling without compromising the utility of visual inputs. We then describe a tactile adapter that injects tactile signals into the pretrained

policy via cross attention, enabling tactile-aware behaviors with minimal additional parameters.

### 3.1. Problem Formulation

The goal of our robot policy is to predict an action chunk based on current multimodal observations. We therefore model a conditional distribution $p(A_t|O_t)$ where $A_t = [a_t, a_{t+1}, \cdots, a_{t+H-1}]$, $t$ is the current time step and $H$ is the length of the action chunk, $O_t = [I_{1,t}, \cdots, I_{n,t}, q_t, T_t, c]$ is the current multimodal information, $I_{i,t}$ is the $i$-th image from one of two active binocular camera views at time $t$ and $q_t \in \mathbb{R}^{28}$ is the robot's current joint states (14 arm joints, 12 hand joints, and 2 active-camera joints). Each action $a_t \in \mathbb{R}^{28}$ is the corresponding target joint positions. $T_t$ is the tactile information, $c$ is the one-hot task condition for each subtask.

During training, we supervise these action tokens using a rectified flow-matching loss.

$$\mathcal{L}_\theta = E_{\tau \sim p(\tau), A, O_t} \left[ ||v_\theta(A_t^\tau, \tau, O) - (\epsilon - A_t^0)||^2 \right] \quad (1)$$

where $\tau$ denotes the flow matching timestep, $p(\tau)$ is the distribution of $\tau$, $A_t^\tau = (1 - \tau) * A_t^0 + \tau * \epsilon$ is the noised action sequence and $A_t^0$ is the original action chunk at time $t$. $\epsilon \sim \mathcal{N}(0, I)$ is Gaussian noise. For each training loop, we sample $\tau = \sigma(\xi) \quad \xi \sim U(0, 1)$ where $\sigma(\cdot)$ is the sigmoid function.

During inference, we generate actions by sampling a series of flow matching timesteps $[\tau_1, \tau_2, \cdots, \tau_k, \tau_{k+1}]$ and apply velocity $v_\theta(A_t^{\tau_i}, \tau_i, O_t)$ at each $\tau_i$, where $k$ is the number of inference steps, $\tau_1 = 1, \tau_{k+1} = 0$.

$$A_t^{\tau_{i+1}} = A_t^{\tau_i} + v_\theta(A_t^{\tau_i}, \tau_i, O_t)(\tau_{i+1} - \tau_i) \quad (2)$$

where $A_t^{\tau_1} = A_t^1 \sim \mathcal{N}(0, I)$ is the initial Gaussian noise.

### 3.2. Multimodal Diffusion Transformer Block

The velocity predictor is built upon our Multi-Modal Diffusion Transformer (MMDiT) Block, as illustrated in Figure 3. The core design principle of MMDiT is to decouple modality-specific conditioning from the attention structure, enabling flexible integration of heterogeneous sensory inputs. We hypothesize that different sensing modalities contribute unequally to action generation and should therefore influence the policy through distinct mechanisms.

Among all modalities, vision plays a dominant role in both human manipulation and modern robotic policies. Accordingly, MMDiT adopts joint self-attention between visual tokens and action tokens, allowing visual information to directly guide action generation. Specifically, binocular images are encoded using a shared ResNet-34 backbone. The resulting feature maps are flattened into token sequences,

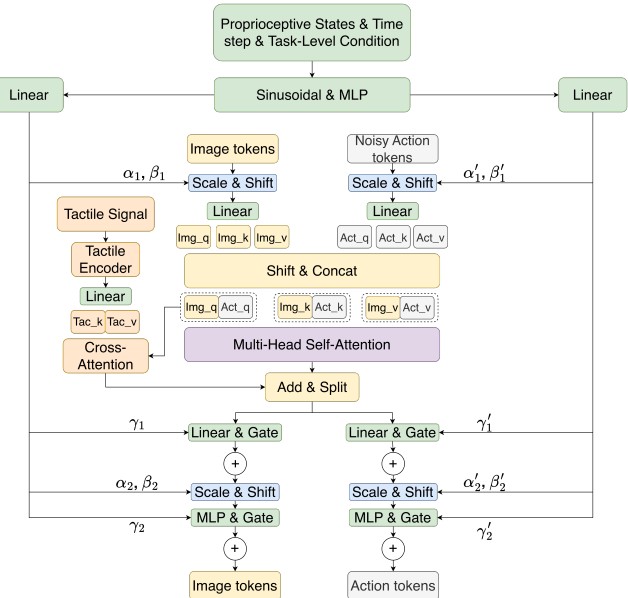

*Figure 3.* **Multimodal Diffusion Transformer Block with Decoupled Conditioning.** Images via self-attention, proprioceptive states via AdaLN, and tactile signals via cross-attention, enabling independent and efficient integration of each modality.

to which rotary positional embeddings (RoPE) (Su et al., 2024) are applied independently. The tokens from the left and right views are then concatenated to form the visual token sequence. In parallel, the noisy action sequence is embedded into action tokens and is augmented with a learnable position embedding. Visual tokens and action tokens are separately projected through linear layers to obtain modality-specific query, key, and value representations. These $QKV$ tensors are then concatenated to construct a unified attention space. After applying root mean square normalization, self-attention is computed over the combined representation, and the outputs are subsequently split back into their respective modalities.

To incorporate additional conditioning signals while preserving the attention structure, both visual tokens and action tokens are further processed by modality-specific linear projections followed by AdaLN (Peebles & Xie, 2023) modulation. The AdaLN parameters are conditioned on proprioceptive states $q_t$, diffusion timesteps $\tau_i$, and optional task-level conditions $c$. The modality-wise conditioning parameters are generated in a AdaLN manner:

$$\alpha_i = f_\theta(x_t) \quad \gamma_i = g_\theta(x_t) \quad \beta_i = h_\theta(x_t) \quad (3)$$

where $f_\theta$, $g_\theta$ and $h_\theta$ are 2-layer MLPs with SiLU activation, and $x_t$ is the addition of proprioceptive embeddings, task condition embeddings, and flow matching timesteps embeddings at times $t$. These parameters are applied to

action tokens and image tokens in the MMDIT block by scale, shift and gate operations as illustrated in Figure 3.

In addition, tactile signals are incorporated through a dedicated cross-attention module, enabling lightweight and plug-and-play integration of tactile sensing without modifying the self-attention structure of the pretrained vision-based policy. Finally, the separated action tokens are processed by a linear prediction head to estimate the action velocity in the diffusion process.

### 3.3. Tactile Adapter for Vision-Based Policy

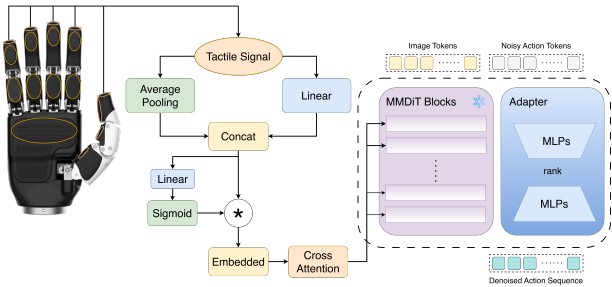

*Figure 4.* **Plugin Tactile Adapter.** Raw tactile information is encoded by the tactile encoder and integrated into the pretrained policy via LoRA for efficient adaptation.

To incorporate tactile signals into a pretrained vision-based DECO, we introduce a plug-in tactile adapter for parameter-efficient tactile conditioning without modifying pretrained parameters. The adapter contains a tactile encoder and cross-attention modules for injecting tactile information, while Low Rank Adaptation(LoRA) (Hu et al., 2021) selectively fine-tunes the attention layers of the pretrained vision–action backbone. During the second training stage, the pretrained policy is frozen, and only the adapter parameters are optimized, enabling tactile-aware behaviors with minimal additional capacity.

As shown in Figure 4, the tactile encoder produces region-level features using two complementary approaches: averaging raw values within each tactile pad (covering finger tips, pulps, ends, and palm) and projecting raw values through a learnable linear layer. Features $T$ from both approaches are concatenated, and a gating mechanism controls their relative influence to produce the final tactile embeddings $e_t$.

$$e_t = \text{MLP}\Big(\text{Sinusoid}(\sigma(W \cdot T) \cdot T)\Big), \qquad (4)$$

where $\sigma$ denotes the $sigmoid$ function and $W$ represents a learnable linear layer. These tactile embeddings are injected into the pretrained vision-based DECO via cross-attention, where the LoRA-based adapter modulates the attention projections in a low-rank manner. This design enables the model to selectively attend to tactile cues that are relevant to contact-rich interactions, such as pressing and assembly,

while maintaining the original visual manipulation capabilities learned during pretraining.

## 4. Experiments

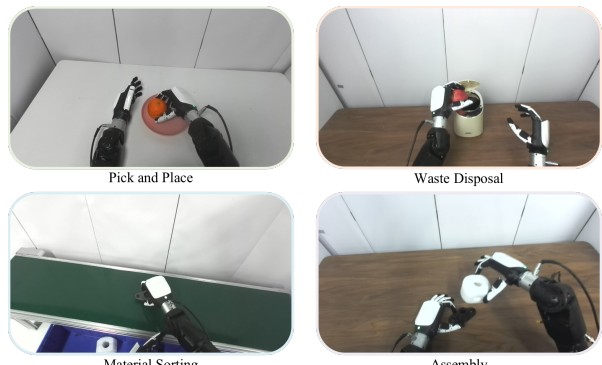

*Figure 5.* **Task Illustration.** DECO-50 dataset comprises four scenarios, each consisting of multiple sub-tasks.

We collect a dataset containing four **bimanual dexterous** manipulation tasks with multimodal data: active binocular images, **tactile** signals, and robot proprioception. We train our method on this dataset with and without tactile data. Our experiments aim to answer the following questions:

- Do all tasks need tactile to achieve high performance?

- Can tactile adapter improve visual-based policies?

- How to integrate tactile data into multimodal policies?

### 4.1. Dataset

As shown in Figure 5, the dataset comprises four tasks, each designed to evaluate different aspects and levels of bimanual dexterous manipulation.

**Task 1: Pick and Place.** The robot moves a plate with its left hand while picking up objects from the table and placing them on the plate with its right hand. Although static pick-and-place is simple, the dual-arm setting introduces higher-dimensional action spaces and increases complexity. This task evaluates basic bimanual coordination.

**Task 2: Material Sorting.** The robot grasps moving objects on a conveyor belt and places them into the corresponding containers with the appropriate hand. This task introduces dynamic robot–object interaction and evaluates the policy's visual precision and reaction speed.

**Task 3: Waste Disposal.** The robot uses one hand to pick up and throw trash into a bin while using the other to open and close the bin by pressing. Visual feedback is less informative when opening or closing the lid. The task thus evaluates the benefit of tactile sensing in contact-rich phases.

*Table 1.* Policy Performance on Four Tasks.

| Method | Pick and Place | Material Sorting | Waste Disposal | Assembly | Success Rate 1[3] | Success Rate 2[4] |
|---|---|---|---|---|---|---|
| ACT | 101/120 | 97/120 | 21/80 | 10/80 | 57.25% | 19.38% |
| DP | 93/120 | 67/120 | 30/80 | 15/80 | 51.25% | 28.13% |
| DP.t[1] | 97/120 | 77/120 | 38/80 | 15/80 | 56.75% | 33.13% |
| DECO | 103/120 | 101/120 | 48/80 | 37/80 | 72.25% (↑ 21.00%) | 53.13% (↑ 33.75%) |
| DECO.p[2] | **108/120** | **105/120** | **62/80** | **55/80** | **82.50% (↑ 31.25%)** | **73.13% (↑ 53.75%)** |

[1] DP.t: Concatenates tactile information into the input and is trained from scratch.
[2] DECO.p: Integrates tactile modality with the tactile adapter.
[3] Success Rate 1: Average success rate on all tasks.
[4] Success Rate 2: Average success rate on contact-rich tasks, which are Waste Disposal and Assembly.

**Task 4: Assembly.** The robot simultaneously controls a socket and a plug with both hands to complete assembly. This contact-rich task requires more precise bimanual coordination and force control than waste disposal.

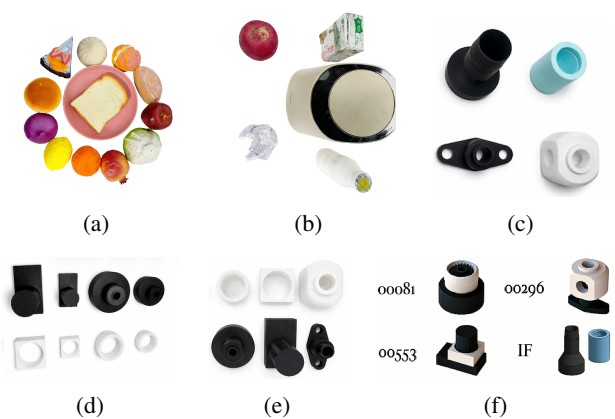

(a)  (b)  (c)

00081  00296

00553  IF

(d)  (e)  (f)

*Figure 6.* Objects used in (a) Pick-and-Place, (b) Waste Disposal, (c) ablation experiments, (d) Assembly, (e) Material Sorting, and (f) the piece–correspondence mapping. The four-digit codes (e.g., 00081, 00553) are object identifiers from the Automate Dataset (Tang et al., 2024); postfixes "-1.5x", "-2x", "-3x" indicate the physical print scale. IF denotes a custom interference-fit pair with a soft hose and a rigid nozzle.

Objects used in all experiments are shown in Figure 6. The data collection process is described in Appendix A.3. For more details about the dataset please refer to Appendix A.

### 4.2. Experiment Setup

**Baselines.** We use ACT (Zhao et al., 2023) and DP (Chi et al., 2025) as visual-only baselines. ACT, DP, and our model are first trained from scratch without tactile data. To evaluate the effect of tactile information, we then create two variants: DP.t, in which tactile data are simply concatenated into the input and the model is trained from scratch, and DECO.p, where the tactile adapter is integrated into our

pretrained vision-based policy and fine-tuned.

**Real-World Setting.** We follow the same setup as in data collection to ensure consistency between training and deployment. The only differences is a slight and unavoidable variations in the relative pose between the robot and the table, which naturally occur during real-world execution.

### 4.3. Result and Analysis

Table 1 summarizes the overall performance across all four tasks. Per-task results are reported in Tables 2–4, where the four-digit object IDs and postfixes (e.g., "-2x") follow the naming in Figure 6f. Detailed success counts with Wilson 95% confidence intervals are provided in Appendix B.2.

**Task 1 (Pick and Place).** All methods perform well because objects are static and grasp/release can be handled with visual and proprioceptive feedback alone, yet DECO still outperforms the baselines. This likely benefits from its architecture, which processes visual information via joint self-attention with action tokens rather than DP's simple FiLM-based (Perez et al., 2018) conditioning. Results also show that adding tactile yields only limited gains, indicating that vision and proprioception are sufficient for this task. Detailed results for Task 1 are provided in Appendix B.2.

*Table 2.* Performance on Material Sorting Task

| Object | ACT | DP | DP.t | DECO | DECO.p |
|---|---|---|---|---|---|
| 00081-s[1]2x | **17/20** | 13/20 | 12/20 | 16/20 | 17/20 |
| 00081-p[2]2x | 6/20 | 5/20 | 12/20 | 10/20 | **13/20** |
| 00296-s-2x | 16/20 | 11/20 | 10/20 | **17/20** | **17/20** |
| 00296-p-2x | **20/20** | 11/20 | 10/20 | 18/20 | 18/20 |
| 00553-s-3x | 19/20 | 18/20 | 18/20 | **20/20** | **20/20** |
| 00553-p-3x | 19/20 | 9/20 | 15/20 | **20/20** | **20/20** |
| Total | 97/120 | 67/120 | 77/120 | 101/120 | **105/120** |

[1] s: S represents the black socket in the pair.
[2] p: P represents the white plug in the pair.

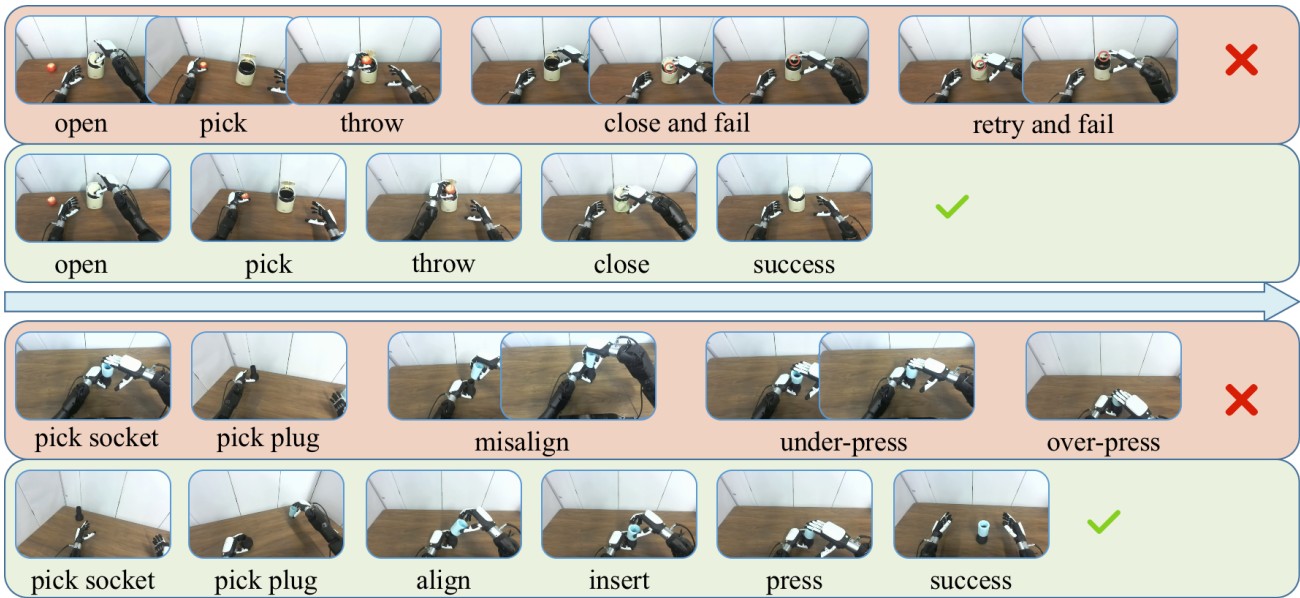

*Figure 7.* DECO with and without tactile on Waste Disposal and Assembly tasks

**Task 2 (Material Sorting).** DP performs relatively poorly compared to ACT and our model, likely due to slower inference, which hinders reliable grasping of objects moving on the conveyor. Both ACT and DP have the lowest success on 00081-p-2x, the smallest object with circular shape and smooth surface, which tends to slip or bounce when the grasp force is too high or too low. With tactile data, policies can distinguish whether the hand and object are in contact and avoid insufficient or excessive force during grasping. Both DP.t and DECO.p improve notably on 00081-p-2x, and DP.t also improves on 00553-s-3x, another small, challenging object. For the remaining larger objects, tactile brings limited improvement, as vision and proprioception suffice for stable grasping, consistent with Task 1.

compound action that involves not only aligning and sealing the lid but also pressing a button to fully secure it. While vision and proprioception can guide the hand to the contact position, they cannot reliably indicate whether the lid is closing, whereas tactile signal helps detect successful contact and monitor the applied force. As in Figure 7, a vision-only policy may wrongly assume the lid is closed and withdraw the hand, causing the lid to bounce and forcing a retry. With the tactile adapter, our model better distinguishes closed vs. open and improves on Stages 1 and 3 (Table 3). Using tactile in DP and in our model both raise success rates on those stages; ACT does well on Stage 1 but struggles to close the lid without tactile. See Appendix B.2 for more.

*Table 3.* Performance on Waste Disposal Task

| Stage | ACT | DP | DP.t | DECO | DECO.p |
|---|---|---|---|---|---|
| Stage1 | 79/80 | 67/80 | 72/80 | 71/80 | 76/80 |
| Stage2 | 66/80 | 38/80 | 65/80 | 65/80 | 76/80 |
| Stage3 | 21/80 | 30/80 | 48/80 | 48/80 | **62/80** |
| Results | 21/80 | 30/80 | 38/80 | 48/80 | **62/80** |

*Table 4.* Performance on Assembly Task

| Object | ACT | DP | DP.t | DECO | DECO.p |
|---|---|---|---|---|---|
| 00081-2x | 8/20 | 8/20 | 5/20 | 15/20 | **18/20** |
| 00081-1.5x | 0/20 | 1/20 | 3/20 | 3/20 | **10/20** |
| 00553-3x | 2/20 | 4/20 | 7/20 | 10/20 | **14/20** |
| 00553-2x | 0/20 | 2/20 | 0/20 | 9/20 | **13/20** |
| Total | 10/80 | 15/80 | 15/80 | 37/80 | **55/80** |

**Task 3 (Waste Disposal).** This longer-horizon task has two strongly tactile-dependent phases: opening and closing the bin lid. We report success counts up to each of three stages: Stage 1 (open lid), Stage 2 (pick and throw trash), and Stage 3 (close lid). Successfully opening or closing the lid requires applying appropriate torque, which depends on both the applied force and the moment arm determined by the contact position. In particular, closing the lid is a

**Task 4 (Assembly).** Assembly is the most challenging task. We evaluate all policies on 4 pairs of objects, as shown in Table 4, using the object assets from (Tang et al., 2024) scaled to different sizes. Both DP and DECO improve when using tactile data, while ACT struggles without tactile signal. We observe that all models perform better on larger objects, mainly because smaller objects are harder to pinch and more prone to visual occlusion during both grasping

and assembly. Tactile signal helps policies detect whether contact has been established during picking and maintain a stable plug–socket configuration during insertion, where interaction forces between the two parts can otherwise cause slipping or dropping. See Appendix B.2 for details.

**Summary.** Across the four tasks, we can distinguish weakly vs. strongly tactile-relevant settings. Pick-and-place and material sorting are weakly tactile-relevant: vision and proprioception usually suffice, so the extra cost of tactile collection may not be needed. In contrast, waste disposal and assembly are strongly tactile-relevant tasks where tactile signal is essential for key subgoals such as detecting contact, monitoring applied forces, and handling visual occlusion. Adding tactile significantly improves both DP and DECO on these tasks. Our plugin tactile adapter effectively incorporates tactile information into pretrained vision-based policies with minimal overhead, requiring far fewer parameters and less training time than training from scratch.

### 4.4. Ablation Studies

As shown in Section 4.3, our plugin tactile adapter brings significant gains to visual-based DECO in complex contact-rich tasks. To further assess its effectiveness, we perform ablation on two representative assembly object pairs (Figure 6c, Table 5) and train two DECO variants from scratch with tactile: DECO.cs utilizes tactile in a coupled way, simply adding tactile embeddings to the mixed conditioning described in Eq. 3, and DECO.ds uses the same preprocessing and cross-attention described in Section 3.3 but is trained from scratch. The assembly of object 00296-2x follows the stages of the main assembly task. The interference-fit (IF) pair is an additional pair with a soft hose and a rigid nozzle, whose stages are: Stage 1 (pick up both parts), Stage 2 (align the hose with the nozzle and slide it on), and Stage 3 (press the hose past the raised retention ring to achieve a firm interference fit without bottoming out).

For the first pair, unlike the sockets and plugs in Task 4, the plug is larger than the socket, which causes severe visual occlusion during the final assembly stage. In addition, the black socket is small and smooth, making it difficult to pinch stably without blocking the insertion trajectory of the plug. When vision can no longer observe the local contact state, tactile signal provides complementary cues across stages to indicate successful insertion. The second pair also requires tactile signal to disambiguate different stages. As illustrated in Figure 7, without tactile DECO is prone to misalignment, under-pressing (failing to pass the retention ring), or over-pressing (bottoming out and stressing the hose). With tactile signal, DECO can sense whether the hose has slid past the retention ring and use force cues during pressing as reliable signals for completion, improving success rates particularly on Stage 2 and Stage 3.

*Table 5.* Ablation Studies on Assembly Tasks

| Object | Stage | DECO | DECO.cs[1] | DECO.ds[2] | DECO.p |
|---|---|---|---|---|---|
| 00296-2x | 1 | 20/20 | 18/20 | 19/20 | 17/20 |
| | 2 | 17/20 | 17/20 | 17/20 | 17/20 |
| | 3 | 10/20 | 10/20 | **14/20** | 13/20 |
| IF | 1 | 13/20 | 15/20 | 18/20 | 19/20 |
| | 2 | 11/20 | 15/20 | 17/20 | 16/20 |
| | 3 | 8/20 | 11/20 | 15/20 | **16/20** |
| Total | 1 | 33/40 | 33/40 | 37/40 | 36/40 |
| | 2 | 28/40 | 32/40 | 34/40 | 33/40 |
| | 3 | 18/40 | 21/40 | **29/40** | **29/40** |

[1] DECO.cs: Tactile is injected in a **coupled** way with proprioception and trained from **scratch**.
[2] DECO.ds: Tactile is injected in a **decoupled** way with other modality and trained from **scratch**.

From Table 5, DECO.ds and DECO.p consistently outperform the baseline DECO on both object pairs, whereas DECO.cs behaves similarly to DECO. This indicates that simply appending tactile embeddings (DECO.cs) is insufficient, while our tactile adapter (DECO.p / DECO.ds) can effectively extract and exploit tactile information in assembly. This is reasonable because tactile signals are sparse in both time (short contact intervals) and space (only a few pads are activated) and are further corrupted by sensor noise; naive fusion can make it difficult for the policy to isolate informative tactile patterns.

Overall, DECO.p (plugin adapter) achieves performance comparable to DECO.ds (trained from scratch) on both pairs, despite updating far fewer parameters. The gap between DECO and DECO.cs highlights the importance of how tactile is injected, and the gains from DECO to DECO.ds/DECO.p demonstrate that our tactile preprocessing and cross-modal attention mechanism enables substantially better use of tactile data than plain conditioning merges, confirming that tactile information is crucial for complex contact-rich assembly tasks.

As shown in Table 6, DECO.p has a comparable total model size to DECO.ds, with the difference coming solely from the LoRA modules. Despite this, DECO.p requires updating only 7.97M trainable parameters (tactile encoder, cross-attention key/value projections for tactile, and LoRA) while achieving comparable performance to DECO.ds trained from scratch, demonstrating that tactile information can be incorporated efficiently without retraining the entire model.

A core principle of DECO is that different modalities should be injected through dedicated pathways. To verify that the specific modality-to-pathway assignment matters, we swap

*Table 6.* The parameters of DECO and its variants.

| Parameters(M) | DECO | DECO.cs | DECO.ds | DECO.p |
|---|---|---|---|---|
| Total | 83.05 | 84.14 | 89.41 | 91.02 |
| Trainable | 83.05 | 84.14 | 89.41 | **7.97** |

the injection mechanisms of the two auxiliary modalities: **proprioceptive** states via cross-attention and **tactile** signals via AdaLN, while keeping all other components identical to DECO.ds.

*Table 7.* Ablation on Conditioning Pathway Assignment

| Object | Stage | DECO.ds | DECO.ds (permuted)[1] |
|---|---|---|---|
| | 1 | 19/20 | 15/20 |
| 00296-2x | 2 | 17/20 | 6/20 |
| | 3 | **14/20** | 0/20 |
| | 1 | 18/20 | 18/20 |
| IF | 2 | 17/20 | 11/20 |
| | 3 | **15/20** | 0/20 |
| | 1 | 37/40 | 33/40 |
| Total | 2 | 34/40 | 17/40 |
| | 3 | **29/40** | 0/40 |

[1] DECO.ds (permuted): proprioception via cross-attention, tactile via AdaLN.

As shown in Table 7, swapping the pathways causes a dramatic drop, especially on Stage 3 (0/40 vs. 29/40). Real-world experiments exhibit noticeable jitter, as injecting proprioceptive inputs via cross-attention treats each joint as a sequence element and causes the model to overemphasize proprioceptive signals. Conversely, AdaLN captures more global conditioning, which is less effective for the fine-grained tactile sensitivity required in assembly. These results confirm that the specific modality-to-pathway assignment in DECO is crucial.

## 5. Conclusion

In this paper, we present DECO, a decoupled multimodal Diffusion Transformer paradigm for bimanual dexterous manipulation that separately conditions on visual, proprioceptive, and tactile modalities. We also introduce a plugin tactile adapter that efficiently incorporates tactile information into pretrained DECO models with minimal parameter overhead. Alongside our model, we release DECO-50, a large-scale bimanual dexterous manipulation dataset with tactile information and active vision in the real world, which will serve as a valuable resource for bimanual dexterous manipulation and tactile-based policy research.

Our experiments on DECO-50 demonstrate that tactile infor-

mation benefits primarily contact-rich tasks (waste disposal and assembly), where it enables contact detection, force monitoring, and handling of visual occlusion. In contrast, simpler tasks (pick-and-place and material sorting) achieve strong performance with vision and proprioception alone, suggesting that tactile collection may not be justified for such scenarios. The plugin tactile adapter brings significant improvement on complex contact-rich tasks while requiring fewer than 10% of the model parameters and minimal training time compared to training from scratch.

While DECO achieves strong performance across most tasks, several limitations remain. The current evaluation is restricted to a single hardware platform with one type of dexterous hand and tactile sensor. Future work will extend DECO along multiple directions: validating the tactile adapter on larger-scale pretrained VLAs and across diverse datasets, hardware platforms, and dexterous hands with varying tactile sensing modalities; and incorporating temporal modeling and memory mechanisms to better handle long-horizon contact-rich manipulation.

## Acknowledgements

We sincerely appreciate the dedication and effort of HongZhi Du, WenZhuo Li, YuXin Liu, GuangLiang Sun, PengBo Sun, XinRui Wang, and XueNing Zhu for their invaluable assistance with data collection, quality assurance, and real-world testing throughout this project.

## Impact Statement

This paper presents work whose goal is to advance the field of Machine Learning. There are many potential societal consequences of our work, none which we feel must be specifically highlighted here.

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

# Supplementary Material

## A. Dataset Details

### A.1. Dataset Size

*Table 8.* Detailed success statistics and total collected hours for the four tasks.

| Task | Object | Succ./Total Traj. | Succ./Total Hours |
|---|---|---|---|
| Task 1 | Onion | 132/160 | 1.001/1.267 |
| | Apple | 113/131 | 1.003/1.168 |
| | Lemon | 144/160 | 1.005/1.113 |
| | Orange | 123/128 | 1.001/1.044 |
| | Pomegranate | 141/150 | 1.029/1.109 |
| | Bread | 160/175 | 1.033/1.142 |
| | Cream Puff | 156/165 | 1.036/1.101 |
| | Hamburger | 178/189 | 1.043/1.112 |
| | Cabbage | 237/298 | 1.340/1.756 |
| | Cake | 172/190 | 1.013/1.142 |
| | Steamed Bun | 181/196 | 1.083/1.174 |
| | Bread Slice | 159/178 | 0.997/1.138 |
| Task 2 | 00081 | 797/1521 | 3.300/5.253 |
| | 00553 | 837/1009 | 3.528/4.085 |
| | 00296 | 1091/1944 | 3.532/5.939 |
| Task 3 | Paper Ball | 300/358 | 2.502/2.965 |
| | Plastic Bottle | 300/337 | 2.317/2.593 |
| | Rotten Apple | 300/320 | 2.080/2.260 |
| | Expired Milk | 300/358 | 2.259/2.775 |
| Task 4 | 00081 1.5x | 301/375 | 2.195/2.755 |
| | 00081 2x | 353/457 | 2.401/3.004 |
| | 00553 2x | 300/334 | 2.165/2.344 |
| | 00553 3x | 318/325 | 1.953/1.993 |
| | 00296 2x | 575/697 | 4.340/5.234 |
| | Interference-Fit | 353/467 | 3.548/4.561 |

### A.2. Objects used in the dataset

*Table 9.* Physical Material Reproduction

| Automate Material ID | 00081 | 00553 | 00296 |
|---|---|---|---|
| **Material Sorting Task** | 2x | 3x | 2x |
| **Assembly Task** | 2x&1.5x | 3x&2x | 2x |

The materials used in the material sorting and assembly tasks can be reproduced by 3D printing. All parts are printed with a Bambu Lab H2D printer, with some geometries scaled up from their original sizes following (Tang et al., 2024); details are given in Table 9. Additionally, we design a custom pair of interference-fit (IF) parts for the ablation study. All components are printed in black or white Bambu Lab PLA Basic, except for the soft hose in the custom IF pair, which is printed in TPU.

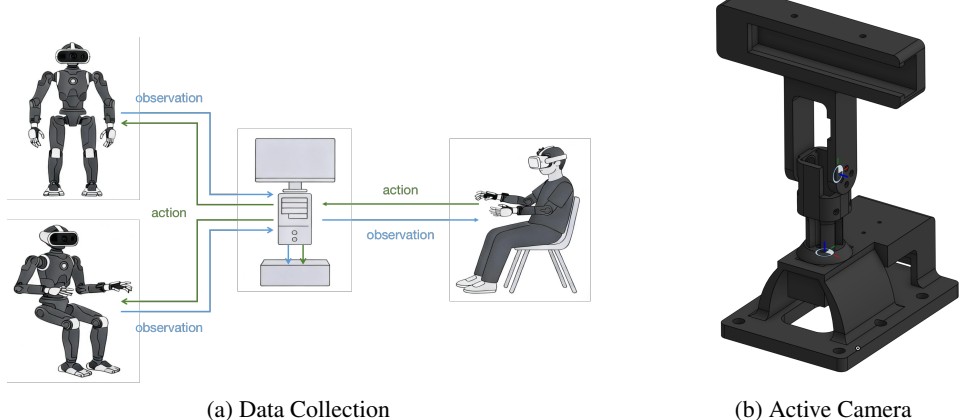

(a) Data Collection        (b) Active Camera

*Figure 8.* Data collection setup with the active camera.

## A.3. Data Collection

We set up a bimanual teleoperation system with a custom active camera mounted on Unitree H1-2. The robot is equipped with a pair of Inspire RH56DFTP hands; each hand has 6 DOF and 17 tactile pads (1062 contact points in total, each with values 0–4096). The dual arms have 14 DOF and the active camera has yaw and pitch motors.

*Table 10.* Tactile pad layout of the Inspire RH56DFTP hand (per hand).

| Pad Position | Array Size (rows $\times$ cols) | Taxel Count |
| --- | :---: | ---: |
| Fingertip (5 fingers) | $3 \times 3$ (each) | $5 \times 9 = 45$ |
| Finger top (5 fingers) | $12 \times 8$ (each) | $5 \times 96 = 480$ |
| Finger palm (4 fingers) | $10 \times 8$ | $80 \times 4 = 320$ |
| Thumb middle | $3 \times 3$ | 9 |
| Thumb palm | $12 \times 8$ | 96 |
| Hand palm | $8 \times 14$ | 112 |
| **Total per hand** | — | 1062 |

The piezoresistive tactile sensors measure only **normal force** (pressure) and do not capture shear or tangential force components. Table 10 details the array size of each tactile pad. During manipulation, the most frequently engaged regions are the finger tops and the hand palm, which provide the densest spatial sampling. In waste disposal tasks, sustained contact with the lid produces stable normal-force signals that indicate whether sufficient force is applied. In assembly tasks, misalignment between the socket and plug shifts the contact region across tactile pads, providing cues for pose adjustment.

During teleoperation, we control only the upper body from the human's motion. The lower body is set to a motion mode when the robot must stand or a damping mode when it sits. In both cases, the torso joints are fixed so that the upper body is aligned with the torso. Human motion is captured by a Vision Pro, and wrist pose and hand keypoints are sent to a PC, which runs inverse kinematics to obtain arm commands and retargets the human hand to the robot hand. The PC acts as a bidirectional bridge: it sends these commands to the robot and, in turn, receives binocular images, tactile readings, and proprioception from the robot, records them, and streams the first-person view back to the Vision Pro. States and actions are logged at 30 Hz.

## A.4. Dataset Comparison

*Table 11.* Dataset Comparison (✔: yes, ✘: no, ✔/✘: partial/mixed, –: not reported)

| Dataset | Bimanual | Dexterous | Tactile | Real | Hours | Episodes |
|---|---|---|---|---|---|---|
| DROID (Khazatsky et al., 2024) | ✘ | ✘ | ✘ | ✔ | 350 | 76K |
| RH20T (Fang et al., 2024) | ✘ | ✘ | ✘ | ✔ | – | 110K |
| BRMData (Zhang et al., 2024) | ✔/✘ | ✘ | ✘ | ✔ | – | 500 |
| RoboMIND (Wu et al., 2025a) | ✔/✘ | ✔/✘ | ✔/✘ | ✔/✘ | 305.5 | 107K |
| RoboMIND2.0 (Hou et al., 2025) | ✔ | ✔/✘ | ✔/✘ | ✔/✘ | 1000+ | 310K |
| RoboCOIN (Wu et al., 2025c) | ✔ | ✔/✘ | ✘ | ✔ | – | 180K |
| RoboTwin (Mu et al., 2025) | ✔ | ✘ | ✘ | ✘ | – | – |
| RoboTwin2.0 (Chen et al., 2025) | ✔ | ✘ | ✘ | ✘ | – | 100k |
| REASSEMBLE (Sliwowski et al., 2025) | ✘ | ✘ | ✘ | ✔ | – | 4551 |
| Touch100k (Cheng et al., 2024) | ✘ | ✘ | ✔ | ✔/✘ | – | – |
| FreeTacMan (Wu et al., 2025b) | ✘ | ✘ | ✔ | ✔ | – | 10k |
| VTDexManip (Liu et al., 2024) | ✔/✘ | ✔ | ✔ | ✔/✘ | – | – |
| Open X-Embodiment (O'Neill et al., 2024) | ✔/✘ | ✘ | ✘ | ✔ | – | 1M+ |
| BridgeData V2 (Walke et al., 2024) | ✘ | ✘ | ✘ | ✔ | – | 53.9k |
| UniHand-2.0 (Being-H0.5) (Luo et al., 2026) | ✔/✘ | ✔/✘ | ✘ | ✔/✘ | 35000+ | – |
| DECO-50 | ✔ | ✔ | ✔ | ✔ | 50 | 8K |

BRMData includes some bimanual data but provides neither dexterous-hand nor tactile modalities; RoboMIND 1.0 contains bimanual dexterous-hand data but does not mention tactile sensing; RoboMIND 2.0 includes bimanual dexterous-hand data as well as bimanual gripper-based tactile information; RoboCOIN provides bimanual dexterous-hand data; Open X-Embodiment covers bimanual robot data across embodiments; FreeTacMan provides gripper-oriented tactile data; and VTDexManip contains bimanual dexterous-hand visuo-tactile data.

## B. Experiment Details

### B.1. Training Details

We train the above models independently on each of four tasks using 8 NVIDIA A100 GPUs(40GB). For each task, we compute the mean and variance of observations independently to standardize the inputs. In particular, images are resized to $256 \times 256$ using a long-side resizing method to preserve aspect ratios. Additionally, for Task 1, Task 3, and Task 4, we incorporate a task-level one-hot condition into the model to distinguish different objects and improve generalization. All models are optimized using AdamW with betas=(0.95, 0.999), weight_decay=1e-6, and an initial learning rate of 1e-4 decayed to 1e-6 using a cosine schedule. The batch size is set to 2048.

We re-implement ACT and DP by closely following the official LeRobot repository. To ensure a comparable model capacity across different baselines, we modify the default channel configuration of the conditional U-Net used in DP from [512, 1024, 2048] to [256, 512, 1024], resulting in a similar parameter scale to our model.

For DP.t, tactile observations are encoded using several linear layers and are directly concatenated with visual and proprioceptive features as input to the policy, without any modality-specific decoupling.

For DECO, The MMDiT backbone consists of 6 transformer blocks with a hidden dimension of 512. The tactile encoder is a 2-layer MLP ($1062 \times 2 \rightarrow 17 \times 2$, with Mish activation) that maps the concatenated raw taxel readings of both hands to region-level features. These learned features are concatenated with the average-pooled region features (17 per hand), yielding a 68-dimensional vector for bimanual input. A linear gate ($\sigma(W \cdot x)$ with $W \in \mathbb{R}^{68 \times 68}$, no bias) modulates the concatenated features, followed by a learnable positional embedding that projects each region into the transformer hidden dimension for cross-attention. The LoRA rank is set to 32. All hyperparameters and training configurations are provided in the open-sourced code at `github.com/BAAI-Humanoid/DECO`.

For all models, after training for a predefined number of epochs (see Table 12 for details), we select the best-performing

*Table 12.* Training Details

| Hyperparameter | ACT | DP | DP.t | DECO | DECO.p |
|---|---|---|---|---|---|
| Optimizer | AdamW | AdamW | AdamW | AdamW | AdamW |
| Betas | [0.95, 0.999] | [0.95, 0.999] | [0.95, 0.999] | [0.95, 0.999] | [0.95, 0.999] |
| Weight Decay | 1e-6 | 1e-6 | 1e-6 | 1e-6 | 1e-6 |
| Batchsize | 2048 | 2048 | 2048 | 2048 | 2048 |
| Learning Rate | 1e-4 | 1e-4 | 1e-4 | 1e-4 | 1e-4 |
| ModelSize | 51.60M | 76.46M | 78.30M | 83.05M | 91.02M |
| Trainable Parameters | 51.60M | 76.46M | 78.30M | 83.05M | **7.97M** |
| Traning Epochs | 30 | 200 | 200 | 150 | 150 |
| Image Encoder | ResNet18 | ResNet18 | ResNet18 | ResNet34 | ResNet34 |
| Action Chunk Size | 32 | 32 | 32 | 32 | 32 |
| Execution Chunk Size | 1 | 16 | 16 | 16 | 16 |
| Sampler | — | DDIM | DDIM | Flow Matching | Flow Matching |
| Inference Steps | — | 10 | 10 | 5 | 5 |
| Temporal Ensembler | ✔ | — | — | — | — |

*Table 13.* Task1 Pick and Place Details

| Object | ACT | DP | DP.t | DECO | DECO.p |
|---|---|---|---|---|---|
| Onion | 2/10 | **9/10** | 8/10 | 8/10 | 7/10 |
| Apple | **10/10** | 6/10 | 9/10 | 9/10 | 8/10 |
| Lemon | 9/10 | **10/10** | 9/10 | 8/10 | 9/10 |
| Orange | **10/10** | 7/10 | 7/10 | 9/10 | 9/10 |
| Pomegranate | **10/10** | 9/10 | 8/10 | 9/10 | **10/10** |
| Bread | 8/10 | 9/10 | 8/10 | 8/10 | **10/10** |
| Cream Puff | 8/10 | **10/10** | 8/10 | 9/10 | 9/10 |
| Hamburger | **10/10** | 9/10 | 7/10 | **10/10** | **10/10** |
| Cabbage | **10/10** | 3/10 | **10/10** | **10/10** | 9/10 |
| Cake | 7/10 | 6/10 | 7/10 | 7/10 | **10/10** |
| Steamed Bun | 7/10 | **10/10** | 7/10 | 9/10 | **10/10** |
| Bread Slice | **10/10** | 5/10 | 9/10 | 7/10 | 7/10 |
| Total | 101/120 [76.6%, 89.6%] | 93/120 [69.2%, 84.1%] | 97/120 [72.9%, 86.9%] | 103/120 [78.5%, 91.0%] | **108/120 [83.3%, 94.2%]** |

checkpoint based on validation performance and deploy it for real-world evaluation.

## B.2. Detailed Experiment Results

We show detailed experimental results on each sub-object in all four tasks. Values in square brackets in the Total rows denote Wilson 95% confidence intervals (in percent).

**Task 1 (Pick-and-Place).** We evaluate all methods on 12 objects in the pick-and-place setting, reporting per-object success counts out of 10 trials. As shown in Table 13, performance varies substantially across objects, reflecting different grasping difficulty and execution robustness. The last row summarizes the overall success across all 120 trials.

*Table 14.* Task2 Material Sorting Details

| Object | ACT | DP | DP.t | DECO | DECO.p |
|---|---|---|---|---|---|
| 00081-Socket-2x | **17/20** | 13/20 | 12/20 | 16/20 | **17/20** |
| 00081-Plug-2x | 6/20 | 5/20 | 12/20 | 10/20 | **13/20** |
| 00296-Socket-2x | 16/20 | 11/20 | 10/20 | **17/20** | **17/20** |
| 00296-Plug-2x | **20/20** | 11/20 | 10/20 | 18/20 | 18/20 |
| 00553-Socket-3x | 19/20 | 18/20 | 18/20 | **20/20** | **20/20** |
| 00553-Plug-3x | 19/20 | 9/20 | 15/20 | **20/20** | **20/20** |
| 00081-Socket-2x (OOD) | –/20 | –/20 | –/20 | 11/20 | **13/20** |
| 00081-Plug-2x (OOD) | –/20 | –/20 | –/20 | 15/20 | **19/20** |
| Black Mouse (OOD) | –/20 | –/20 | –/20 | 4/20 | **18/20** |
| Total | 97/120 [72.9%, 86.9%] | 67/120 [46.9%, 64.4%] | 77/120 [55.3%, 72.2%] | 101/120 [76.6%, 89.6%] | **105/120 [80.4%, 92.3%]** |

**Task 2 (Material Sorting).** We evaluate fine-grained part sorting on three part families (00081/00296/00553), each containing socket and plug variants, with 20 trials per category. Table 14 reports detailed success counts and additionally includes out-of-distribution (OOD) objects to test generalization. Overall, the OOD results reveal a clear performance drop for most methods, highlighting the challenge of robust recognition and sorting under distribution shifts.

*Table 15.* Task3 Waste Disposal Details

| Object | Stage | ACT | DP | DP.t | DECO | DECO.p |
|---|---|---|---|---|---|---|
| Paper Ball | Stage1 | **20/20** | 16/20 | 18/20 | 15/20 | 16/20 |
| | Stage2 | **16/20** | 12/20 | 8/20 | 14/20 | **16/20** |
| | Stage3 | 11/20 | 7/20 | 8/20 | 11/20 | **12/20** |
| Plastic Bottle | Stage1 | **20/20** | 14/20 | **20/20** | 19/20 | **20/20** |
| | Stage2 | 15/20 | 5/20 | 14/20 | 18/20 | **19/20** |
| | Stage3 | 4/20 | 5/20 | 9/20 | 11/20 | **14/20** |
| Rotten Apple | Stage1 | 19/20 | 18/20 | **20/20** | 19/20 | **20/20** |
| | Stage2 | 19/20 | 6/20 | 8/20 | 19/20 | **20/20** |
| | Stage3 | 3/20 | 6/20 | 8/20 | 12/20 | **20/20** |
| Expired Milk | Stage1 | **20/20** | 19/20 | 19/20 | 18/20 | **20/20** |
| | Stage2 | 16/20 | 15/20 | 13/20 | 14/20 | **17/20** |
| | Stage3 | 3/20 | 12/20 | 13/20 | 14/20 | **16/20** |
| Total | Stage1 | **79/80 [93.3%, 99.8%]** | 67/80 [74.2%, 90.3%] | 77/80 [89.5%, 98.7%] | 71/80 [80.0%, 94.0%] | 76/80 [87.8%, 98.0%] |
| | Stage2 | 66/80 [72.7%, 89.3%] | 38/80 [36.9%, 58.3%] | 43/80 [42.9%, 64.3%] | 65/80 [71.3%, 88.3%] | **72/80 [81.5%, 94.8%]** |
| | Stage3 | 21/80 [17.9%, 36.8%] | 30/80 [27.7%, 48.5%] | 38/80 [36.9%, 58.3%] | 48/80 [49.0%, 70.0%] | **62/80 [67.2%, 85.3%]** |

**Task 3 (Waste Disposal).** Table 15 reports stage-wise success statistics on four objects (20 trials per stage). Across all methods, performance generally decreases from Stage 1 to Stage 3, indicating increasing difficulty and accumulated execution errors in the multi-stage throwing procedure. Notably, our method achieves substantially higher success in Stage 3 overall, suggesting improved robustness to object dynamics and long-horizon uncertainties.

*Table 16.* Task4 Assembly Details

| Object | Stage | ACT | DP | DP.t | DECO | DECO.p |
|---|---|---|---|---|---|---|
| 00081-2x | Stage1 | **20/20** | **20/20** | **20/20** | **20/20** | 19/20 |
| | Stage2 | **20/20** | 15/20 | 8/20 | 16/20 | 19/20 |
| | Stage3 | 8/20 | 8/20 | 5/20 | 15/20 | **18/20** |
| 00081-1.5x | Stage1 | 9/20 | 19/20 | **20/20** | 17/20 | 17/20 |
| | Stage2 | 0/20 | **14/20** | 10/20 | 10/20 | 13/20 |
| | Stage3 | 0/20 | 1/20 | 3/20 | 3/20 | **10/20** |
| 00553-3x | Stage1 | **20/20** | 16/20 | 18/20 | 17/20 | **20/20** |
| | Stage2 | **19/20** | 9/20 | 14/20 | 15/20 | 16/20 |
| | Stage3 | 2/20 | 4/20 | 7/20 | 10/20 | **14/20** |
| 00553-2x | Stage1 | 6/20 | **19/20** | 18/20 | 14/20 | 14/20 |
| | Stage2 | 0/20 | 6/20 | 9/20 | **14/20** | **14/20** |
| | Stage3 | 0/20 | 2/20 | 0/20 | 9/20 | **13/20** |
| Total | Stage1 | 55/80 [57.9%, 77.8%] | 74/80 [84.6%, 96.5%] | **76/80 [87.8%, 98.0%]** | 70/80 [78.5%, 93.1%] | 70/80 [78.5%, 93.1%] |
| | Stage2 | 39/80 [38.1%, 59.5%] | 44/80 [44.1%, 65.4%] | 41/80 [40.5%, 61.9%] | 55/80 [57.9%, 77.8%] | **63/80 [68.6%, 86.3%]** |
| | Stage3 | 10/80 [6.9%, 21.5%] | 15/80 [11.7%, 28.7%] | 15/80 [11.7%, 28.7%] | 37/80 [35.7%, 57.1%] | **55/80 [57.9%, 77.8%]** |

**Task 4 (Assembly).** Table 16 summarizes detailed results for the assembly task under multiple settings, including automated execution with different difficulty scales (e.g., 1.5x/2x/3x) and interference fit parts. These settings stress precise alignment and contact-rich interaction, where small pose errors can lead to failure. Overall, the detailed breakdown helps reveal method robustness under stricter tolerances and distribution shifts.

### B.3. Supplemental Experiments with ResNet-18 Vision Encoder

Since ACT and DP both use ResNet-18 as their default vision encoder, we replace DECO's default ResNet-34 backbone with ResNet-18 for both DECO and DECO.p to isolate the effect of encoder capacity on policy performance. After the replacement, DECO (ResNet-18) has 72.94M parameters and DECO.p (ResNet-18) has 80.91M parameters, which is lower than DP (76.46M) and higher than ACT (51.60M). As shown in Tables 17–20, the performance difference between ResNet-18 and ResNet-34 is marginal across all tasks. Despite the reduced parameter count, DECO (ResNet-18) still consistently outperforms both baselines, confirming that the decoupled conditioning pathway design, rather than encoder capacity, is the primary driver of the performance gains.

*Table 17.* ResNet-18 vs. ResNet-34 on Task 1 (Pick and Place)

| Object | DECO (ResNet-34) | DECO (ResNet-18) | DECO.p (ResNet-34) | DECO.p (ResNet-18) |
|---|---|---|---|---|
| Onion | 8/10 | 8/10 | 7/10 | **10/10** |
| Apple | 9/10 | **10/10** | 8/10 | 9/10 |
| Lemon | 8/10 | **10/10** | 9/10 | 9/10 |
| Orange | 9/10 | **10/10** | 9/10 | **10/10** |
| Pomegranate | 9/10 | 8/10 | **10/10** | **10/10** |
| Bread | 8/10 | 9/10 | **10/10** | **10/10** |
| Cream Puff | 9/10 | **10/10** | 9/10 | **10/10** |
| Hamburger | **10/10** | **10/10** | **10/10** | **10/10** |
| Cabbage | **10/10** | 9/10 | 9/10 | 9/10 |
| Cake | 7/10 | **10/10** | **10/10** | **10/10** |
| Steamed Bun | 9/10 | 9/10 | **10/10** | **10/10** |
| Bread Slice | 7/10 | 8/10 | 7/10 | **9/10** |
| Total | 103/120 [78.5%, 91.0%] | 111/120 [86.4%, 96.0%] | 108/120 [83.3%, 94.2%] | **116/120 [91.7%, 98.7%]** |

*Table 18.* ResNet-18 vs. ResNet-34 on Task 2 (Material Sorting)

| Object | DECO (ResNet-34) | DECO (ResNet-18) | DECO.p (ResNet-34) | DECO.p (ResNet-18) |
|---|---|---|---|---|
| 00081-s-2x | 16/20 | 18/20 | 17/20 | **19/20** |
| 00081-p-2x | 10/20 | 14/20 | 13/20 | **15/20** |
| 00296-s-2x | **17/20** | 14/20 | **17/20** | 15/20 |
| 00296-p-2x | 18/20 | **19/20** | 18/20 | 16/20 |
| 00553-s-3x | **20/20** | **20/20** | **20/20** | **20/20** |
| 00553-p-3x | **20/20** | 19/20 | **20/20** | 18/20 |
| Total | 101/120 [76.6%, 89.6%] | 104/120 [79.4%, 91.6%] | **105/120 [80.4%, 92.3%]** | 103/120 [78.5%, 91.0%] |

*Table 19.* ResNet-18 vs. ResNet-34 on Task 3 (Waste Disposal)

| Object | Stage | DECO (ResNet-34) | DECO (ResNet-18) | DECO.p (ResNet-34) | DECO.p (ResNet-18) |
|---|---|---|---|---|---|
| Paper Ball | 1 | 15/20 | 19/20 | 16/20 | **20/20** |
| | 2 | 14/20 | 16/20 | 16/20 | **19/20** |
| | 3 | 11/20 | 11/20 | 12/20 | **14/20** |
| Plastic Bottle | 1 | 19/20 | 18/20 | **20/20** | **20/20** |
| | 2 | 18/20 | 13/20 | **19/20** | 17/20 |
| | 3 | 11/20 | 10/20 | **14/20** | 12/20 |
| Rotten Apple | 1 | 19/20 | **20/20** | **20/20** | **20/20** |
| | 2 | 19/20 | **20/20** | **20/20** | 19/20 |
| | 3 | 12/20 | 18/20 | **20/20** | 17/20 |
| Expired Milk | 1 | 18/20 | 17/20 | **20/20** | **20/20** |
| | 2 | 14/20 | 12/20 | **17/20** | 16/20 |
| | 3 | 14/20 | 11/20 | **16/20** | 14/20 |
| Total | 1 | 71/80 [80.0%, 94.0%] | 74/80 [84.6%, 96.5%] | 76/80 [87.8%, 98.0%] | **80/80 [95.4%, 100.0%]** |
| | 2 | 65/80 [71.3%, 88.3%] | 61/80 [65.9%, 84.2%] | **72/80 [81.5%, 94.8%]** | 71/80 [80.0%, 94.0%] |
| | 3 | 48/80 [49.0%, 70.0%] | 50/80 [51.5%, 72.3%] | **62/80 [67.2%, 85.3%]** | 57/80 [60.5%, 80.0%] |

*Table 20.* ResNet-18 vs. ResNet-34 on Task 4 (Assembly)

| Object | Stage | DECO (ResNet-34) | DECO (ResNet-18) | DECO.p (ResNet-34) | DECO.p (ResNet-18) |
|---|---|---|---|---|---|
| 00081-2x | 1 | **20/20** | **20/20** | 19/20 | **20/20** |
| | 2 | 16/20 | **19/20** | **19/20** | 18/20 |
| | 3 | 15/20 | 14/20 | **18/20** | **18/20** |
| 00081-1.5x | 1 | 17/20 | 16/20 | 17/20 | **18/20** |
| | 2 | 10/20 | 14/20 | 13/20 | **15/20** |
| | 3 | 3/20 | 11/20 | 10/20 | **13/20** |
| 00553-3x | 1 | 17/20 | 19/20 | **20/20** | 17/20 |
| | 2 | 15/20 | 7/20 | **16/20** | 12/20 |
| | 3 | 10/20 | 1/20 | **14/20** | 9/20 |
| 00553-2x | 1 | 14/20 | **18/20** | 14/20 | 17/20 |
| | 2 | **14/20** | 13/20 | **14/20** | **14/20** |
| | 3 | 9/20 | 11/20 | 13/20 | **14/20** |
| Total | 1 | 70/80 [78.5%, 93.1%] | **73/80 [83.0%, 95.7%]** | 70/80 [78.5%, 93.1%] | 72/80 [81.5%, 94.8%] |
| | 2 | 55/80 [57.9%, 77.8%] | 53/80 [55.4%, 75.7%] | **63/80 [68.6%, 86.3%]** | 59/80 [63.2%, 82.1%] |
| | 3 | 37/80 [35.7%, 57.1%] | 37/80 [35.7%, 57.1%] | **55/80 [57.9%, 77.8%]** | 54/80 [56.6%, 76.8%] |

