# OpenReview forum: "DECO: Decoupled Multimodal Diffusion Transformer for Bimanual Dexterous Manipulation with a Plugin Tactile Adapter"
_ICML.cc/2026/Conference — ICML 2026 regular_

### Official Review · Reviewer_74bS · 2026-03-04

**Soundness:** 3
**Presentation:** 3
**Significance:** 4
**Originality:** 3
**Overall Recommendation:** 5
**Confidence:** 3

**Summary:**

- Proposes a Diffusion Transformer policy that handles through specialized conditioning pathways vision, proprioception and tactile inputs for bimanual manipulation tasks.
- Uses a tactile adapter to inject tactile information into pretrained DECO
- Introduces DECO-50, a dataset of 50h of data for bimanual dexterous manipulation with tactile sensing (4 scenarios, 28 sub-tasks)
Real-world experiments shows DECO with tactile adaptation achieves better success rate than vision-only baselines

**Compliance With Llm Reviewing Policy:**

Affirmed.

**Final Justification:**

Thanks for the clarifications to my questions. I have updated my score.

**Key Questions For Authors:**

- Could the authors provide the operational specifications of the tactile sensors and and an analysis of the specific contact events they are best suited to capture?
- How does the tactile signal variance compare across different tasks in DECO-50?
Is the tactile encoder trained from scratch, or is it a pretrained model fine-tuned using LoRA?
- How does the architecture handle the inherent sparsity of tactile data compared to the continuous, dense nature of the vision and action modalities?
- Is there an analysis (e.g., saliency maps or perturbation studies) to characterize which tactile features the model prioritizes—binary contact, normal force variations, or shear forces?
- Why does tactile information significantly improve performance in Task 3 (Stage 2) but show negligible impact in Task 1, despite the tasks appearing qualitatively similar?
- What is the success rate for a full execution of Task 3 without breaking it down into individual sub-tasks?
- “Pick-and-place and material sorting are weakly tactile-relevant” Is this a general claim, or a limitation of the current experimental setup (e.g., Fig. 6 suggests objects differ in geometry but share similar material properties)?

**Limitations:**

yes

**Strengths And Weaknesses:**

__Strengths__

- Addresses the critical challenge of effectively integrating tactile modalities into Diffusion Transformer policies.
- The chosen baselines are representative of the current state-of-the-art, providing a fair benchmark for the proposed method.
- The 4 tasks incorporated in the dataset are interesting from a tactile perspective

__Weaknesses__

- Tactile adapter has been previously explored in [1]. It’s suggested to compare the advantages of the proposed approach.
- The manuscript lacks technical details regarding the tactile sensors, specifically their spatial resolution and the dynamic range of contact events they are capable of capturing
- The data collection method (teleoperation) often lacks intentional force modulation. It remains unclear if the recorded tactile signals provide sufficient information for the policy to learn closed-loop force control rather than simple contact detection

[1] Higuera, C., Sharma, A., Fan, T., Bodduluri, C. K., Boots, B., Kaess, M., ... & Mukadam, M. (2025, October). Tactile beyond pixels: Multisensory touch representations for robot manipulation. In Conference on Robot Learning (pp. 105-123). PMLR.

---

> ### Author Rebuttal · Authors · 2026-03-31
>
> We sincerely thank you for the helpful questions.
>
> **Q1. Tactile sensor specifications and supported contact events.**
> Each Inspire RH56DFTP hand has **1062 taxels**. The pad sizes are 3×3 (fingertip), 12×8 (finger top), 10×8 (four-finger palm), 3×3 (thumb middle), 12×8 (thumb palm), and 8×14 (hand palm), with force resolution **0.1 N**. These sensors measure **normal force / pressure only**, not shear. Therefore, they are better suited to **static or quasi-static contact patterns** than slip-dominated events. This matches our main tactile-sensitive settings: **Task 3** (sustained pressing on the lid) and **Task 4** (force-sensitive alignment).
>
> **Q2. Tactile signal variance across tasks; tactile encoder training.**
> We currently provide a **qualitative** comparison of tactile variation. In **Task 1/2**, the main tactile changes occur during grasp / release. In **Task 3**, strong tactile variation appears during lid contact and sustained pressing. In **Task 4**, tactile variation arises during both grasping and force-sensitive alignment.
> The **tactile encoder is always trained from scratch**. It is not pretrained.
>
> **Q3. How do we handle tactile sparsity?**
> We address this through conditioning pathway design.
> Simply **concatenating** sparse tactile features with dense vision features suppresses tactile information, as evidenced by the performance gap between DECO.cs and DECO.ds. We instead use cross-attention to inject tactile signals, which provides a stronger mechanism for sparse tactile data to influence the vision-action feature stream. To validate this design, we conduct a **pathway-swapping ablation** (As listed in **Reviewer Hptr Q4**)—injecting proprioception via cross-attention and tactile via AdaLN—and observe noticeable jitter as well as insufficient pressing in the IF assembly stage, confirming the design is essential.
>
> **Q4. Which tactile features does the model prioritize?**
> We don't yet include a dedicated saliency or perturbation analysis, so we do not claim to have fully characterized the model’s reliance on specific tactile features. However, our results suggest the policy uses more than binary contact detection: in **Task 3** and the **interference-fit** ablation, success depends on distinguishing **insufficient vs. sufficient pressing**, not just contact/no-contact. Since our sensors do not measure shear explicitly, we also do **not** claim to resolve shear-specific effects.
>
> **Q5. Why does Task 3 Stage 2 appear to improve, while Task 1 shows limited tactile gain?**
> The increase in Stage 2 success counts in Task 3 is primarily driven by the improvement in Stage 1. To make this clearer, we supplement the stage 1-to-stage 2 conditional success rate below:
> | | | | | |
> | - | - | - | - | - |
> | | DP | DP.t | DECO | DECO.p |
> | Task1 Success Rate | 93/120=77.5%  | 97/120=80.8%  | 103/120=85.8% | 108/120=90.0% |
> | Task3 Stage 1 to 2 Conditional Success Rate | 38/67 = 56.7% | 43/77 = 55.8% | 65/71 = 91.5% | 72/76 = 94.7% |
>
> Task 1 involves a longer bimanual coordination horizon, whereas Task 3's Stage 1→2 only requires grasping and placing, which partly explains why DECO and DECO.p show a slight improvement. DP and DP.t, however, drop substantially, as their action jitter is tolerated by the large open surface of the plate but causes frequent failures when placing into the narrow opening of the trash bin.
>
> **Q6. Detailed Success Rate for Task 3.**
> Due to page limits, we only report per-stage success rates to highlight the progressive influence of tactile information across task phases; the full per-object, per-stage breakdown is provided in Appendix Table 13.
>
> **Q7. “Weakly tactile-relevant” tasks.**
> We clarify that this is not a general claim but a task-specific observation based on the marginal tactile gains in our experiments. In Task 1, neither soft nor hard objects show notable improvement, suggesting limited tactile relevance under our setup. We acknowledge the lack of systematic material variation in Task 2/3/4 and will address this in future work.
> | | | | | | |
> |---|---|---|---|---|---|
> ||ACT|DP|DP.t|DECO|DECO.p|
> |Soft Object|51/60|44/60|51/60|53/60|52/60|
> |Hard Objecrt|50/60|49/60|46/60|50/60|56/60|
>
> **W1. Comparison with *Tactile Beyond Pixels*.**
> We thank the reviewer for this relevant reference; TBP makes valuable contributions to large-scale multisensory touch representation learning. Architecturally, TBP adopts a ControlNet-style adapter that duplicates the original network blocks, whereas we use LoRA-based adaptation; for feature fusion, TBP adds tactile features element-wise, while we inject them via cross-attention. Beyond these architectural differences, TBP primarily focuses on multisensory representation, whereas our work centers on how multimodal signals are effectively injected into action generation policies.
>
> **W2/W3. Sensor details and force-related interpretation.**
> We will add details from Q1 to the revision.
>
> Thank you again for your comments.

---

> > ### Author Rebuttal · Reviewer_74bS · 2026-04-01
> >
> > Thanks for the clarifications to my questions.

---

### Official Review · Reviewer_v2wB · 2026-03-11

**Soundness:** 2
**Presentation:** 3
**Significance:** 2
**Originality:** 2
**Overall Recommendation:** 3
**Confidence:** 3

**Summary:**

The submission proposes DECO, a multimodal diffusion transformer with a parameter-efficient tactile adapter. On a set of challenging real-world manipulation tasks, DECO performs better than vision-only baselines and baselines in which tactile data are concatenated into the input. A tactile bimanual dataset, DECO-50, is also released.

**Compliance With Llm Reviewing Policy:**

Affirmed.

**Final Justification:**

The rebuttal reinforced my prior assessment that the proposed adapter is not deployable in most scenarios of interest. The comparison against naive touch concatenation is too weak a baseline in my opinion. I will keep my score.

**Key Questions For Authors:**

Why didn't you compare DECO against established alternatives that use more sophisticated techniques for fusing tactile signals (e.g. ForceVLA)?

Why didn’t you perform adapter training and fine-tuning (DECO.p) on a different dataset to that used to pretrain the vision-action backbone (beyond simply adding tactile signals)?

**Limitations:**

yes

**Strengths And Weaknesses:**

Presentation:

The submission is clearly written and well structured, with a good literature review and informative figures.

Soundness:

The set of tasks is good. In particular, the waste disposal and assembly tasks have a strong tactile component.

DECO is compared against a bespoke tactile-aware baseline (DP.t) in which tactile data are concatenated into the input. It is not compared against any established baseline(s) that use more sophisticated techniques for fusing tactile signals. This is a significant weakness of the submission, as it makes it difficult to assess how DECO compares to other tactile-aware methods.

The way in which the tactile adapter is tested seems misaligned with how it would be used in practice. If a dataset has tactile signals, presumably one would train DECO on the whole dataset from scratch (i.e. use DECO.ds, not DECO.p). If the vision–action backbone had already been pretrained on a dataset without tactile signals, and subsequently a different dataset with tactile signals was available, DECO.p would seem appropriate. However, fine-tuning a model on a different dataset is not examined here, and may present different challenges.

The improved performance of DECO.ds compared to DECO.cs nicely demonstrates the benefits of the cross-attention-based adapter architecture versus simply adding tactile embeddings to the mixed conditioning.

Originality:

The individual components of DECO are not original. Cross-attention-based adapters have been used before in other contexts (e.g. Flamingo). Parameter Efficient Fine-Tuning (PERT) with methods such as LoRA is well established.

Significance:

The authors address an important and timely problem - the incorporation of touch sensing into diffusion models. While the results appear promising and clearly show the benefits of incorporating touch, the lack of a strong tactile-aware baseline makes it difficult to know how much DECO advances the field. In addition, the tactile adapter appears to have been tested in a setting different from how it would be used in practice, limiting what conclusions can be drawn about its usefulness as a post-training adapter.

The companion dataset will be useful to the community.

---

> ### Author Rebuttal · Authors · 2026-03-31
>
> We sincerely thank you for the thoughtful review. We appreciate both of your core concerns: (1) the absence of comparison against stronger tactile-aware baselines, and (2) the fact that the current DECO.p experiment is not yet a true cross-dataset post-training adapter evaluation.
>
> **Q1. Why did we not compare against established tactile-aware baselines (e.g., ForceVLA)?**
> We agree that comparison against stronger tactile-aware baselines would strengthen the paper. The main reason we did not include such a comparison is that most existing tactile/force-aware methods differ substantially from our setting in one or more of the following dimensions: **embodiment** (single-arm/gripper vs. bimanual dexterous hands), **sensor modality** (wrist F/T, visual-tactile, or sparse tactile vs. dense hand-mounted tactile arrays), **action space/control interface**, and **model scale/training regime**. As one concrete example, ForceVLA uses 6-axis wrist force/torque sensing, gripper-based manipulation, and a much larger pi0-based VLA backbone, whereas DECO operates on dense tactile arrays on bimanual dexterous hands with a lightweight diffusion policy. Because of these differences, a direct apples-to-apples comparison would be difficult to interpret and would not cleanly isolate the effect of tactile-fusion design.
>
> For this reason, we chose a controlled within-setting evaluation:
> - **DECO.cs vs. DECO.ds** isolates the effect of tactile injection design (coupled vs. decoupled).
> - **DECO.ds vs. DECO.p** isolates the effect of the plugin adapter paradigm (full retraining vs. parameter-efficient adaptation).
> - **DECO vs. DP.t** tests whether naive tactile concatenation is sufficient.
>
> We agree that this does **not** establish superiority over the broader tactile-aware literature. Rather, our current evidence supports a narrower claim: **within the same hardware / data / backbone family, decoupled tactile injection is more effective than coupled or naive fusion**. We will clarify this scope more explicitly in the revision.
>
> **Q2. Why did we not test DECO.p in a different-dataset post-training setting?**
> We agree that the present DECO.p experiment is a **controlled same-dataset proof-of-concept**, rather than the strongest cross-dataset post-training adapter evaluation envisioned in the review. We chose this setup to isolate the effect of parameter-efficient tactile injection without conflating it with dataset shift, embodiment shift, and action-space mismatch.
>
> A true cross-dataset adapter study is important, but it is non-trivial in our current setting because existing manipulation datasets often differ in:
> 1. **Action space** and **Embodiment / kinematics**,  Existing manipulation datasets are typically tied to specific hardware configurations with different kinematic structures. The action space representation varies fundamentally across platforms — joint positions (as in DECO), end-effector poses, or delta action. Recent cross-embodiment generalization work such as Being-H0.5 [2] has demonstrated the importance of unified action space design for enabling cross-embodiment transfer. Without such a unified action representation, directly transferring a vision-action backbone pretrained on one embodiment to another requires substantial architectural modifications that go beyond simple adapter fine-tuning.
> 2. **Tactile or force modality**
> 3. **Data format** (camera setup, control frequency, observation horizon, annotations).
>
> Therefore, our current DECO.p result should be interpreted as showing that the tactile adapter can be injected into a pretrained policy **within a controlled setting**, not yet as evidence that cross-dataset post-training transfer has been established. We will make this limitation explicit in the revision and present cross-dataset / cross-embodiment adapter evaluation as an important next step.
>
> [2] Luo, H. et al., "Being-H0.5: Scaling Human-Centric Robot Learning for Cross-Embodiment Generalization," arXiv:2601.12993, 2025.
>
> **W1. Originality.**
> We agree that the individual building blocks (cross-attention, AdaLN, and LoRA) are established techniques. Our claimed novelty is not in inventing these components in isolation, but in their **task-specific combination for bimanual dexterous tactile manipulation**: assigning different modalities to distinct conditioning pathways, introducing a plugin tactile adapter for a pretrained vision-action policy, and empirically showing that this specific design outperforms coupled or naive tactile fusion. We will clarify this scope of novelty more explicitly in the revised manuscript.
>
> Thank you again for the constructive feedback. Your comments helped us better define the scope of the current claims and the most important next-step evaluations.

---

> > ### Author Rebuttal · Reviewer_v2wB · 2026-04-04
> >
> > Thank you for your detailed response. The narrower claim seems to significantly reduce the value of the work, as it is fairly well acknowledged by the community that naive concatenation of touch is inadequate. A direct comparison against more sophisticated fusion baselines is needed. The non-trivial nature of cross-dataset adapter evaluation also reinforces my concern that the proposed adapter is not deployable in most scenarios of interest. I will keep my score.

---

> > > ### Author Response · Authors · 2026-04-04
> > >
> > > Thank you for your constructive feedback and careful review of our work. We highly appreciate your insights and would like to clarify several key points to address your remaining concerns.
> > >
> > > **Q1. Why did we not compare against established tactile-aware baselines**
> > >
> > > We did not compare against highly specialized tactile fusion baselines because they rely on heavily customized architectures tailored to their own pipelines. Our goal is not to outperform such specially designed methods, but to propose a lightweight, plug-and-play tactile adapter that can  in principle be flexibly integrated with a pretrained vision-based policy, demonstrating effectiveness and transferability in our evaluated settings.
> > >
> > > **Q2. Cross-dataset adapter evaluation**
> > >
> > > We acknowledge the reviewer’s valid concern regarding the generalizability and deployability of the adapter under cross-dataset and real-world scenarios. In future work, we plan to further validate the scalability of our approach by conducting extensive experiments on cross-dataset adaptation and cross-robot deployment, with the aim to improve its practical applicability across more diverse manipulation settings.
> > >
> > > We highly appreciate your insights and briefly summarize the core contributions of this work:
> > > 1. We propose DECO, a decoupled framework that explicitly disentangles visual, tactile, and proprioceptive states, enabling structured and controllable integration of multimodal sensing for robotic manipulation.
> > > 2. Built upon DECO, we introduce a lightweight, plug-and-play tactile adapter that can be conveniently incorporated into a pretrained vision-based policy. Notably, it achieves performance comparable to training from scratch by fine-tuning less than 10% of the total parameters.
> > > 3. We will publicly release DECO-50, a large-scale dataset for bimanual dexterous manipulation covering 4 scenarios, 28 subtasks, 5 million frames, and 50 hours of synchronized tactile and visual demonstrations.
> > >
> > > Thank you again for the constructive suggestions.

---

### Official Review · Reviewer_Hptr · 2026-03-12

**Soundness:** 2
**Presentation:** 2
**Significance:** 2
**Originality:** 3
**Overall Recommendation:** 4
**Confidence:** 4

**Summary:**

The paper proposes DECO, a Diffusion Transformer-based policy for bimanual dexterous manipulation that integrates visual, proprioceptive, and tactile information through modality-specific conditioning pathways. The central hypothesis is that different input modalities contribute differently to action generation, and that a decoupled architecture better exploits this structure than uniform fusion.

The paper identifies three primary contributions: (1) the DECO framework, which routes each modality through a dedicated conditioning mechanism, using joint self-attention for image and action tokens, adaptive layer normalisation (AdaLN) for proprioceptive states and task conditions, and cross-attention for tactile signals; (2) a plugin tactile adapter that grafts tactile capability onto a frozen pretrained vision-action policy using LoRA, updating fewer than 10% of parameters; (3) DECO-50, a bimanual dexterous manipulation dataset with tactile and active stereo vision, comprising four task categories, 28 sub-tasks, over 50 hours of demonstrations, and approximately 5 million frames collected by teleoperation of a dual dexterous-hand platform.

Training follows a two-stage paradigm. In the first stage, the vision and proprioception branches are trained jointly with flow matching. In the second stage, the pretrained backbone is frozen and only the tactile adapter parameters are optimised. The method is evaluated against ACT and Diffusion Policy baselines in the real world with over 2,000 robot rollouts across the four tasks.

**Compliance With Llm Reviewing Policy:**

Affirmed.

**Final Justification:**

After reading the author rebuttal and the follow-up discussion, I raise my score from 3 (Weak Reject) to 4 (Weak Accept) and my confidence to 4.

The authors provided matched-backbone results across Tasks 1, 2, and 4 showing that DECO with ResNet-18 performs comparably to DECO with ResNet-34, which substantially addresses my main concern about fairness of the headline comparison. The action space definition, the permutation ablation, and the typographical errors were all resolved in the first rebuttal round. I remain conditional on the revision following through on the promised confidence intervals and the full matched-backbone comparison including DECO.p.

**Key Questions For Authors:**

1. What are the exact policy outputs? Are they joint torques, target joint positions, joint velocities, or Cartesian targets? A formal definition is needed to interpret the flow-matching objective and to assess reproducibility.

2. DECO uses ResNet-34 while all baselines use ResNet-18, and DECO also has a higher parameter count. Can you provide either (a) DECO results with ResNet-18 and matched parameter count, or (b) ACT and DP results with ResNet-34? Without this, it is unclear how much of the headline improvement reflects architectural design rather than capacity.

3. Both DECO-50 and the implementation are listed as contributions. Are there plans to release them before or at the time of publication? If not, can you provide sufficient implementation detail for an independent group to reproduce the main results?

4. The assignment of proprioception to AdaLN and tactile to cross-attention is central to the design. Has any ablation been conducted that permutes these assignments, to confirm that the specific choice of mechanism per modality is what drives performance rather than simply the presence of separate pathways?

5. The DECO.p total in Table 4 reads "55/100" while all other entries use 80 trials. Is this a typographical error? Separately, why do DECO.ds and DECO.p have different total parameter counts (89.41M versus 91.02M) if the only stated difference is the training regime?

**Limitations:**

The paper does not include a dedicated limitations section. The authors should explicitly acknowledge the restriction to a single hardware platform and sensor suite, the unclear generalisability to other embodiments, the absence of failure mode analysis, the sensitivity of results to demonstration quality and collection protocol, and the lack of any cross-dataset transfer evaluation.

**Strengths And Weaknesses:**

**Soundness**

The paper addresses a practically important and technically difficult problem. The scale of real-robot evaluation, over 2,000 rollouts across four distinct tasks including multi-stage assembly at varying mechanical tolerances, is larger in scale than what is commonly reported in this literature. The two-stage training strategy is practically well-motivated: it allows a vision-only pretrained policy to acquire tactile capability without retraining from scratch. The ablation results in Table 5 are the most convincing part of the paper. The finding that coupled tactile injection (DECO.cs, where tactile is appended to the AdaLN conditioning channel and the model is trained from scratch) fails to improve over the no-tactile baseline while decoupled cross-attention injection (DECO.ds and DECO.p) succeeds is well-supported. Sparse, noise-corrupted tactile signals are easily dominated by the denser visual stream when all modalities share the same conditioning pathway, so this result is physically interpretable. The plugin adapter achieving near-parity with training from scratch while updating only ~8M of ~91M parameters is also a useful and credible practical result.

However, the fairness of the main comparison in Table 1 is a serious concern. As disclosed in Appendix B.1, DECO uses a ResNet-34 image encoder while ACT and DP use ResNet-18, and DECO also has a higher total parameter count (83M versus 51.6M for ACT and 76.5M for DP). These two confounds make it impossible to attribute the 21-point headline improvement cleanly to the decoupled DiT architecture rather than to increased model capacity. A version of DECO with ResNet-18 and matched parameter count, or equivalently baseline models with ResNet-34, is needed to isolate the architectural contribution. Additionally, no statistical uncertainty is reported anywhere in the paper. Each cell in Tables 1 through 5 is a single point estimate. Given the stochasticity of real-robot experiments and the fact that some inter-condition differences are small (e.g., DECO.ds versus DECO.p in Table 5 often differ by one trial per stage), it is not possible to assess whether reported differences are reliable without confidence intervals or repeated experimental blocks.

**Presentation**

The high-level motivation for the decoupled design is well-articulated, and the experimental narrative across the four tasks is organised and easy to follow. The two-stage training diagram (Figure 2) conveys the overall training procedure clearly.

Several notation and figure issues made it difficult to follow the technical details. The symbol $\tau$ in Equation 1 is not defined immediately on introduction, and since $\tau$ conventionally denotes torque in robotics, I was unsure of its intended meaning until later in the text. The observation $O_t$ includes an image sub-index $I_{i,t}$, but it is not explained whether $i$ indexes multiple cameras, how many cameras are used, or how images are tokenised. The AdaLN sub-networks are described as neural networks without specifying depth, width, or non-linearities. Figures 3 and 4 appear inconsistent in their depiction of the tactile adapter, and I could not resolve the discrepancy from the text alone. Figure 6 references ablation conditions that are not described in the surrounding paragraph. Table 4 contains what appears to be a denominator error: the DECO.p total is listed as "55/100" while all other entries use 80 trials. Finally, the policy output space is never formally defined anywhere in the paper. It is unclear to me whether the policy outputs joint torques, target joint positions, joint velocities, or Cartesian commands, which makes it difficult to interpret the flow-matching objective and to assess reproducibility.

**Significance**

Bimanual dexterous manipulation with tactile sensing is a high-impact setting and DECO-50, providing simultaneous dexterous bimanual trajectories with tactile sensing at this scale, addresses a genuine gap. However, neither the dataset nor the code are made available, and all evaluation is conducted exclusively on the authors' self-collected benchmark. When both the method and the evaluation data are contributions of the same paper and neither is publicly accessible, independent verification is not possible, which substantially limits the practical impact of the dataset contribution claim. The evaluation is also restricted to a single hardware platform and a controlled laboratory environment, with no cross-dataset or cross-embodiment transfer experiment. The dataset description in the main text also lacks critical details: the object naming conventions (four-digit codes with postfixes such as "-2x" or "-3x") are not explained, and the teleoperation protocol, including inter-operator consistency, overall collection success rate, and balance across sub-tasks and object instances, is not described.

**Originality**

The combination of modality-specific conditioning pathways with a LoRA-based plugin tactile adapter for a visuomotor policy is a novel and practically motivated contribution. Individual components such as DiT, AdaLN conditioning, and LoRA are established in the literature, but their combination in this specific multimodal manipulation context is non-trivial and the reasoning behind the design is reasonably well-articulated. The two-stage training strategy is not fundamentally new in the transfer learning sense, but its application to tactile-augmented robot policies is a meaningful extension. To my knowledge, no prior work has proposed a modality-decoupled DiT policy with a plugin tactile adapter for bimanual dexterous manipulation, though the limited architectural transparency (see Presentation concerns) makes novelty difficult to assess in full.

---

> ### Author Rebuttal · Authors · 2026-03-31
>
> We sincerely thank you for the thorough and constructive review. Your comments on fairness, reproducibility, and presentation clarity were very helpful.
>
> **Q1. Exact policy outputs.**
> We apologize for the ambiguity. The policy predicts **28-dimensional target joint positions**: 7 DOF for each arm, 6 DOF for each hand, and 2 DOF for the active camera (yaw/pitch). In the revision, we will formalize this in the problem definition and clarify that each action chunk is a sequence of such target joint-position vectors.
>
> **Q2. Fairness of the main comparison / matched-backbone test.**
> We agree that the ResNet-34 vs. ResNet-18 difference is an important fairness concern. We therefore added a matched-backbone experiment on the most challenging task (Task 4, vision-only stage-1 DECO):
>
> | Model | 00081-2x | 00081-1.5x | 00553-3x | 00553-2x | Total | Params |
> |---|---:|---:|---:|---:|---:|---:|
> | DECO (ResNet-18) | 14/20 | 1/20 | 11/20 | 11/20 | **37/80** | 72.94M |
> | DECO (ResNet-34) | 15/20 | 3/20 | 10/20 | 9/20 | **37/80** | 84.05M |
> | DP (ResNet-18) | 8/20 | 1/20 | 4/20 | 2/20 | **15/80** | 76.49M |
>
> These results suggest that the gains are **not solely explained by encoder capacity**. We agree that a fully matched comparison across all tasks would provide a stronger fairness test, and we will make this limitation explicit in the revision.
>
> **Q3. Code/data availability and reproducibility.**
> We are fully committed to releasing both the **DECO-50 dataset** and the **training/deployment code** upon acceptance. To improve reproducibility during review, we will expand the manuscript/supplement with the missing implementation details you identified, including the policy output space, camera indexing and tokenization, AdaLN sub-network specification, tactile adapter parameterization, and dataset collection protocol.
>
> **Q4. Permuting modality-to-pathway assignments.**
> Yes. We conducted an additional ablation by swapping the assignments, i.e., injecting **proprioception via cross-attention** and **tactile via AdaLN**. On the two assembly ablation pairs, the permuted model performs much worse than the original design:
>
> | Setting | Stage 1 | Stage 2 | Stage 3 |
> |---|---:|---:|---:|
> | DECO.ds (permuted) | 33/40 | 17/40 | 0/40 |
> | DECO.ds (original) | 37/40 | 36/40 | 29/40 |
>
> In real-robot execution, the permuted model also shows noticeable jitter and unstable behavior. One possible explanation is that proprioceptive inputs become over-emphasized when treated as token sequences in cross-attention, while sparse tactile signals are not effectively utilized through AdaLN. We will add this ablation to the revision.
>
> **Q5. Table 4 typo and parameter discrepancy.**
> Thank you for catching both issues. The “**55/100**” entry for DECO.p in Table 4 is a **typographical error**; it should be **55/80**. The parameter gap between **DECO.ds (89.41M)** and **DECO.p (91.02M)** is due to the added **LoRA modules** in DECO.p; the other components remain the same.
>
> **W1. Statistical uncertainty.**
> We fully agree that uncertainty reporting would strengthen the paper. Real-robot evaluation in our setting is expensive and low-throughput (manual reset, object repositioning, and no faithful tactile+bimanual simulator), but we agree this does not replace statistical reporting. In the revision, we will acknowledge this limitation and report binomial/Wilson 95% confidence intervals for the success-rate results where applicable; for very small differences (e.g., 1–2 trials in some stage-wise comparisons), we will also state more clearly that they should be interpreted cautiously.
>
> **W2. Notation and figure clarity.**
> We will revise the paper to address the presentation issues you identified: define **τ** immediately; clarify that the image sub-index refers to the **two views of the active binocular camera**, with ResNet features flattened into token sequences; specify the AdaLN sub-network architecture; clarify that Figures 3 and 4 show the same tactile mechanism at different abstraction levels; and make explicit that Figure 6(c) shows the two object pairs used only in the ablation study.
>
> **W3. Cross-dataset / cross-embodiment transfer.**
> We agree that cross-dataset and cross-embodiment evaluation would substantially strengthen the paper, and we will explicitly acknowledge its absence as a limitation. Our current focus is to isolate the multimodal fusion design itself—whether decoupled, modality-specific conditioning outperforms uniform fusion within a controlled setting. We do not claim that generalization across embodiments has been established here.
>
> **W4. Dataset description.**
> We agree that the dataset description should be more complete. In the revision, we will clarify the object naming convention, the teleoperation protocol and inter-operator consistency, collection success statistics, and the data balance across sub-tasks/object instances.
>
> Thank you again for the constructive suggestions.

---

> > ### Author Rebuttal · Reviewer_Hptr · 2026-04-03
> >
> > I thank the authors for the thorough and constructive rebuttal. Several of my concerns have been addressed satisfactorily.
> > The action space question (Q1) is fully resolved: the policy predicts 28-dimensional target joint positions, and the authors commit to formalizing this in the revision.
> >
> > The conditioning mechanism permutation ablation (Q4) is a convincing addition. The dramatic drop in Stage 3 performance (0/40 for the permuted model versus 29/40 for the original) and the reported jitter provide strong empirical support for the design choices. This concern is resolved.
> >
> > The Table 4 typo and the parameter count discrepancy (Q5) are both explained clearly. These concerns are resolved.
> > Regarding statistical uncertainty (W1), I appreciate the commitment to reporting Wilson 95% confidence intervals in the revision. I consider this concern resolved conditionally on the revision following through.
> >
> > My remaining concern is on Q2, the encoder and parameter fairness. The matched-backbone experiment provided in the rebuttal covers only Task 4 and only the vision-only first stage of DECO. The result shown (DECO ResNet-18: 37/80, DECO ResNet-34: 37/80, DP ResNet-18: 15/80) is informative and encouraging, but it does not cover the full evaluation across all four tasks, and it does not include the tactile-augmented DECO.p variant. Given that Task 4 is the hardest task and may not be representative of all settings, a complete matched comparison remains important to fully substantiate the headline claim. I would ask the authors: can you provide the matched ResNet-18 results for at least Tasks 1 and 2 as well, to confirm that the conclusion holds more broadly?
> >
> > Given the positive elements of the rebuttal, I am willing to raise my overall recommendation from 3 (Weak Reject) to 4 (Weak Accept), conditional on the revision including the promised confidence intervals, the full action space specification, and a transparent discussion of the remaining scope limitations of the matched-backbone comparison.

---

> > > ### Author Response · Authors · 2026-04-04
> > >
> > > We sincerely thank you for the thoughtful follow-up and for the positive update. We are encouraged that Q1, Q4, Q5, and W1 are now largely resolved, and we understand that the remaining main concern is the fairness of the encoder / parameter comparison in Q2.
> > >
> > > We agree with the reviewer that the original ResNet-34 vs. ResNet-18 mismatch introduces a fairness confound, and we appreciate the push to examine this issue more carefully. During the rebuttal window, since all experiments require **full model retraining** and **real-robot deployment and evaluation**, we first prioritized the most challenging task (Task 4), as we believed it would be the most sensitive test of whether encoder capacity was driving the gap. As reported in our rebuttal, the matched-backbone Task 4 result was:
> > > - DECO (ResNet-18): 37/80
> > > - DECO (ResNet-34): 37/80
> > > - DP (ResNet-18): 15/80
> > >
> > > Since then, we have continued running the matched-backbone experiments, and we can now also provide Tasks 1 and 2 under the same ResNet-18 setting for DECO. We have completed full experiments for Task 2.
> > >
> > > | | | | | | | | |
> > > | - | - | - | - | - | - | - | - |
> > > | **Model**       | 00081  Socket  2x | 00081 Plug 2x | 00296 Socket 2x | 00296 Plug 2x | 00553 Socket 3x | 00553 Plug 3x | Total   |
> > > | DECO(ResNet-18) | 18/20                         | 14/20                       | 14/20                         | 19/20                       | 20/20                         | 19/20                       | 104/120 |
> > > | DECO(ResNet-34) | 16/20                         | 10/20                       | 17/20                         | 18/20                       | 20/20                         | 20/20                       | 101/120 |
> > > | DP(ResNet-18)   | 13/20                         | 5/20                        | 11/20                         | 11/20                       | 18/20                         | 9/20                        | 67/120  |
> > >
> > > Due to time constraints, for Task 1 we have conducted a partial evaluation on a representative subset consisting of two hard objects and two soft objects.
> > >
> > > |                 |                     |                           |                    |                           |       |
> > > | --------------- | ------------------- | ------------------------- | ------------------ | ------------------------- | ----- |
> > > | **Model**       | Onion (Hard) | Pomegranate (Hard) | Cake (Soft) | Steamed Bun (Soft) | Total |
> > > | DECO(ResNet-18) | 8/10                | 8/10                      | 10/10              | 9/10                      | 35/40 |
> > > | DECO(ResNet-34) | 8/10                | 9/10                      | 7/10               | 9/10                      | 33/40 |
> > > | DP(ResNet-18)   | 9/10                | 9/10                      | 6/10               | 10/10                     | 34/40 |
> > >
> > > These additional results further support the same trend as Task 4: the observed gains are not solely explained by the ResNet-34 encoder choice. At the same time, we fully agree that the most rigorous presentation is a complete matched-backbone comparison across all tasks, and we will make the current scope limitation explicit in the revision.
> > >
> > > We also want to clarify why we first focused on the vision-only stage-1 comparison. The fairness concern raised in Q2 originates specifically from the visual encoder difference, so our first priority during the rebuttal window was to isolate that factor directly. We agree that a full matched comparison including the tactile-augmented DECO.p variant would be even stronger, and we are continuing those experiments as well for the revision.
> > >
> > > More broadly, our intention was not to create an unfair comparison: for ACT and DP we followed their standard published ResNet-18 configurations, while for DECO we initially used ResNet-34 as our own default vision encoder. In hindsight, we agree that this was not the most rigorous choice for a fairness-sensitive comparison, and we are grateful to the reviewer for pointing this out. The new matched-backbone results have already improved our understanding of the issue, and we will reflect this transparently in the revision.
> > >
> > > As noted in our previous response, we will also follow through on the other promised revisions:
> > > (1) formalizing the full action-space specification.
> > >
> > > (2) reporting Wilson 95% confidence intervals.
> > >
> > > (3) completing the full matched-backbone comparison (including DECO and DECO.p) across all tasks for the final revision.
> > >
> > > (4) explicitly discussing the remaining scope limitations of the matched-backbone comparison.
> > >
> > >
> > > We hope these additional matched-backbone results for Tasks 1 and 2 help further reduce the reviewer’s remaining concern, and we sincerely appreciate the opportunity to strengthen the paper along this direction.

---

### Official Review · Reviewer_9mSS · 2026-03-15

**Soundness:** 4
**Presentation:** 3
**Significance:** 3
**Originality:** 3
**Overall Recommendation:** 5
**Confidence:** 4

**Summary:**

The paper presents a multimodal diffusion transformer for bimanual dexterous manipulation, together with a cross-attention + LoRA based tactile adapter for tactile integration and a new real-world dataset, DECO-50. The method separates how vision, proprioception, and tactile inputs are injected into the policy, rather than fusing them in a single coupled pathway. The paper evaluates the approach on four real-world bimanual tasks and reports that DECO outperforms ACT and Diffusion Policy, while the tactile adapter further improves performance on contact-rich tasks such as waste disposal and assembly with far fewer trainable parameters than full retraining. Overall, the manuscript's key finding pertains to the claim that tactile sensing is most valuable for contact-rich, partially occluded manipulation, and that the way tactile information is injected matters substantially for performance.

**Compliance With Llm Reviewing Policy:**

Affirmed.

**Final Justification:**

Questions raised by the reviewer are adequately addressed.

**Key Questions For Authors:**

1. How do the authors expect the proposed tactile adapter to transfer across embodiments, especially to robots without tactile sensing or with substantially different tactile layouts and dexterous hand morphologies?
2. Since the paper argues that the adapter can be incorporated into pretrained vision-based policies, can the authors comment more concretely on how this would integrate with existing VLA formulations (i.e. pi0, grootn1.6, internvla, etc.) beyond the current DECO setup? If this limitation can be addressed, this could significantly boost the paper from weak accept to accept.

**Limitations:**

Yes

**Strengths And Weaknesses:**

Strengths:
1. The experimental results are fairly convincing that the method improves over the selected baselines, especially on contact-rich tasks, and the parameter-efficient tactile adapter is a practically useful contribution.
2. The release of a real-world bimanual dexterous manipulation dataset with tactile sensing is valuable, since such datasets are still relatively scarce.

Weaknesses:
1. The related work discussion on vision-tactile representation learning feels incomplete. Important prior works and datasets such as Binding Touch to Everything [1], SSVTP [2], TVL [3], ObjectFolder [4], and ObjectFolder 2.0 [5] seem relevant and should likely be discussed more carefully in relation to this paper’s representation and adaptation claims.
2. The paper would benefit from a clearer discussion of how this approach may extend to cross-embodiment settings, especially since future robot platforms may not have tactile sensing or may use different hand designs and different types of tactile sensors.
Although not strictly necessary for acceptance, it would strengthen the paper to discuss more explicitly how the proposed tactile adapter could interface with current VLA-style policy formulations, especially since the paper itself positions multimodal policy integration as an important broader direction.

[1] Yang, F., Feng, C., Chen, Z., Park, H., Wang, D., Dou, Y., Zeng, Z., Chen, X., Gangopadhyay, R., Owens, A., & Wong, A. (2024). Binding touch to everything: Learning unified multimodal tactile representations. In Proceedings of the IEEE/CVF Conference on Computer Vision and Pattern Recognition (pp. 26330–26343).

[2] Kerr, J., Huang, H., Wilcox, A., Hoque, R., Ichnowski, J., Calandra, R., & Goldberg, K. (2023). Self-supervised visuo-tactile pretraining to locate and follow garment features. In Proceedings of Robotics: Science and Systems (RSS).

[3] Fu, L., Datta, G., Huang, H., Panitch, W. C.-H., Drake, J., Ortiz, J., Mukadam, M., Lambeta, M., Calandra, R., & Goldberg, K. (2024). A touch, vision, and language dataset for multimodal alignment. In Proceedings of the 41st International Conference on Machine Learning (ICML).

[4] Gao, R., Chang, Y.-Y., Mall, S., Fei-Fei, L., & Wu, J. (2021). ObjectFolder: A dataset of objects with implicit visual, auditory, and tactile representations. In Proceedings of the 5th Conference on Robot Learning (CoRL).

[5] Gao, R., Si, Z., Chang, Y.-Y., Clarke, S., Bohg, J., Fei-Fei, L., Yuan, W., & Wu, J. (2022). ObjectFolder 2.0: A multisensory object dataset for sim2real transfer. In Proceedings of the IEEE/CVF Conference on Computer Vision and Pattern Recognition (pp. 10598–10608).

---

> ### Author Rebuttal · Authors · 2026-03-31
>
> We sincerely thank you for the constructive feedback and positive assessment. We address the main concerns below.
>
> **Q1. Cross-embodiment transfer of the tactile adapter.**
> The adapter is relatively flexible at the tactile-representation level because it operates on **region-level features** rather than fixed taxel identities. In practice, transferring to a new tactile layout would require redefining the **region partition** and training the tactile encoder/adapter based on new sensor data.
> 1. Region-Level Processing for Compatibility
> The adapter operates on region-level features rather than individual taxel values, where a "region" typically corresponds to a single tactile sensor (e.g., a fingertip sensor pad containing multiple tactile points).
> - Normal-force-only sensors (e.g., piezoresistive, capacitive arrays): Each sensor outputs a pressure value array across its pad.
> - 3D force sensors (e.g., visual-tactile sensors like Tac3D [1] and GelSight [2] , or magnetic tactile sensors [3]): These sensors estimate full 3D force distributions, where each taxel provides 3D force data.
>
> Regardless of whether a region contains scalar pressure values or 3D force vectors, our adapter can processes them uniformly through the same pipeline: Region Level Feature Extraction → Self-Gated → Cross-Attention Fusion.
>
> [1] Zhang et al., "Tac3D: A Novel Vision-based Tactile Sensor for Measuring Forces Distribution," arXiv:2202.06211, 2022.
> [2] Yuan et al., "GelSight: High-Resolution Robot Tactile Sensors for Estimating Geometry and Force," Sensors, 2017.
> [3] Youcan Yan et al., "Soft magnetic skin for super-resolution tactile sensing with force self-decoupling," Sci. Robot.
>
> **Q2. How could the adapter interface with VLA-style policies?**
> 1. Tactile Adapter Integration:
>
> In principle, the tactile adapter is compatible with existing attention-based VLA frameworks.  (e.g., OpenVLA, Pi, GR00T). However, we have not yet experimentally validated this integration, and we will make this limitation explicit and leave it for future work.
>
> For example, in Pi0.5, after the VLM (PaliGemma) and action expert (Gemma) compute their representations with full attention, one can introduce an additional cross-attention layer where tactile features act as keys and values, and the latent features serve as queries. The backbone remains frozen, and only the tactile interaction module is fine-tuned via LoRA.
>
> Similarly, in Fig. 3 of GR00T N1 [4], where VLM latents are fused with the action expert via cross-attention, our method simply appends an additional cross-attention layer to incorporate tactile signals.
> Overall, the tactile adapter is a lightweight, plug-and-play module (cross-attention + LoRA) that injects tactile modality into pretrained vision-based policies with minimal modification.
>
> 2. Limitations:
>
> We build DECO instead of using pretrained VLAs mainly due to the lack of large-scale data for bimanual dexterous manipulation, as existing VLA datasets are dominated by parallel gripper tasks. The current version of DECO-50 dataset also lacks language annotations, which we plan to improve in future work.
> To efficiently validate our idea under limited compute and data, we design DECO as a lightweight testbed. In future work, we will scale to larger datasets to train VLA and world models and further evaluate the scalability of the tactile adapter.
>
> [4] Bjorck, Johan, et al. "Gr00t n1: An open foundation model for generalist humanoid robots." arXiv preprint arXiv:2503.14734, 2025.
>
> **W1. Related work on vision-tactile representation learning.**
> We agree that the related-work discussion should better cover prior visuo-tactile representation learning and multisensory datasets. Binding Touch to Everything, SSVTP, and TVL primarily study visuo-tactile representation learning or cross-modal alignment, while ObjectFolder and ObjectFolder 2.0 provide important multisensory object-level resources and benchmarks.
>
> Our work is complementary: rather than learning a unified tactile representation, we focus on how tactile signals are injected into an action-generating manipulation policy, especially via a parameter-efficient adapter for a pretrained vision-based backbone. We will add this distinction explicitly in the related-work section.
>
> Thank you again for the helpful suggestions. We will revise the paper to clarify these scope boundaries more explicitly.
>
> **W2. Extension of the approach.**
>
> We sincerely thank the reviewer for this constructive suggestion, which has helped us strengthen the paper by prompting a more comprehensive discussion of cross-embodiment transfer and VLA integration. We have addressed these aspects in detail in our responses to Q1 and Q2 above.
>
> Thank you again for the helpful suggestions. We will revise the paper to clarify these scope boundaries more explicitly.

---

> > ### Author Rebuttal · Reviewer_9mSS · 2026-04-03
> >
> > Thanks for the clarification.

---

### Decision · Program_Chairs · 2026-04-30

**Decision:**

Accept (regular)

**Comment:**

This paper introduces DECO, a decoupled multimodal diffusion transformer designed for bimanual dexterous manipulation. By disentangling vision, proprioception, and tactile signals into specialized pathways, the model avoids the common issue of critical tactile information being "drowned out" by visual data. The authors also contribute the DECO-50 dataset, providing 50 hours of high-quality tactile-enabled manipulation data, which fills a significant gap in the field.

The recommendation for Acceptance is based on the following justifications:

First, reviewers praised the architecture’s move away from simple feature concatenation toward a more structured integration of touch. The use of a lightweight, "plug-and-play" tactile adapter was noted as a practical solution for adapting to different sensor configurations without full model retraining.

Second, the DECO-50 dataset was recognized as a major contribution, offering a much-needed benchmark for the community to study force-sensitive tasks. During the rebuttal, the authors successfully addressed concerns regarding technical novelty by clarifying the architectural differences between DECO and prior works, specifically highlighting their use of cross-attention and LoRA-based adaptation.

Finally, supplemental analysis provided by the authors demonstrated that DECO maintains robustness across different material types and force-intensive scenarios. With initial technical concerns resolved, the paper is decided to be Accepted.